# MANIPEVALAGENT: PROMPTABLE AND EFFICIENT EVALUATION FRAMEWORK FOR ROBOTIC MANIPULATION POLICIES

**Yiteng Chen**[1,*]    **Huiping Zhuang**[1,*]    **Wenbo Li**[1,*]
**Shiyi Wang**[1]    **Xiangyu Zhao**[2]    **Qingyao Wu**[1,†]
[1]South China University of Technology    [2]Fudan University
[*]Equal contribution    [†]Corresponding author

## ABSTRACT

In recent years, robotic manipulation policies have made substantial progress. However, evaluating these policies typically requires large-scale sampling in simulation benchmarks, leading to high time costs. Moreover, existing evaluation pipelines are usually fixed, do not account for user needs, and report only a single scalar score, lacking interpretability. In contrast, human experts can quickly form an intuitive impression of a policy's capabilities from just a handful of executions. We therefore propose ManipEvalAgent, an efficient, promptable, and dynamically multi-round evaluation framework for robotic manipulation policies. The framework conducts small-batch, multi-round evaluations and adaptively plans subsequent evaluation steps based on intermediate observations from each round. Via code generation, it constructs tasks and evaluation functions within simulator. By generating evaluation functions and leveraging vision-language models (VLMs) for video understanding, ManipEvalAgent provides user-instruction-centric, fine-grained analysis. Our approach offers three key advantages: (1) efficiency, no need for massive sampling; (2) promptable, planning the evaluation process according to user queries; and (3) interpretability, providing diagnostic text that goes beyond a single score. Across multiple settings, our evaluation method significantly shortens the overall time compared with traditional simulation benchmarks, while reaching conclusions comparable to those from large-scale simulation benchmarks.

## 1 INTRODUCTION

In recent years, robotic manipulation has advanced rapidly, driven by progress in diffusion models (Chi et al., 2023) and breakthroughs in general vision-language-action (VLA) models (Kim et al., 2024; Shukor et al., 2025; Liu et al., 2024), as well as the availability of internet-scale demonstration data (O'Neill et al., 2024); together, these developments have pushed end-to-end capabilities from perception to action and expanded the boundaries of practical applications.

With the progress in robotic manipulation policies, effective evaluation has become increasingly critical for identifying their limitations and directions for improvement. Existing benchmarks (Chen et al., 2025; Liu et al., 2023; James et al., 2020; Mees et al., 2022; Yu et al., 2020) provide standardized environments and task suites, together with unified evaluation pipelines and data resources, laying the groundwork for systematic model comparisons and enabling more comprehensive performance analyses. Nevertheless, prevailing practice largely relies on fixed evaluation pipelines and pre-defined task sets, lacks user input and customization, and is ill-suited to open-ended needs. Meanwhile, static benchmarks typically require exhaustive execution over all predefined tasks and all candidate policies, incurring substantial time and compute costs. More importantly, conclusions are often compressed into single metrics such as success rate, with little diagnosis of *why* and *under what conditions* failures occur, making it difficult to directly guide model iteration and system deployment. Compared with static benchmarks, experienced human evaluators can often form a reliable impression of a robotic manipulation policy's overall competence through small-batch, hands-on interactive trials.

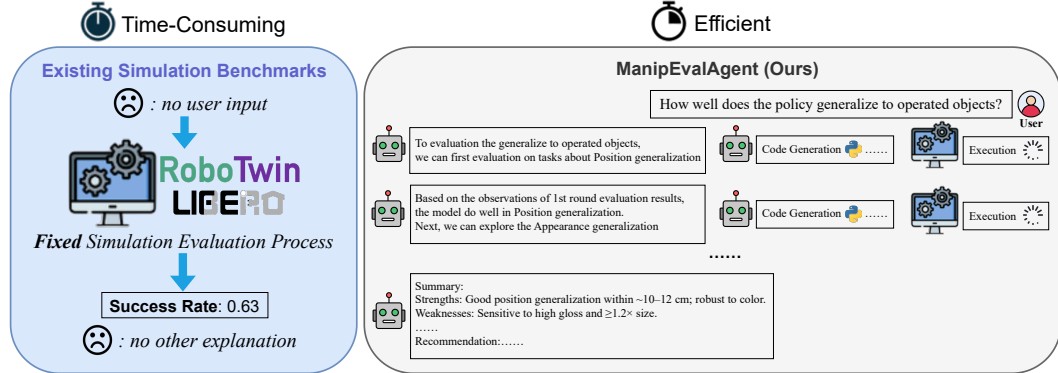

Figure 1: Example of ManipEvalAgent. Widely used simulation benchmarks evaluate on fixed task sets with fixed procedures, require large amounts of sampling, and report only success rates. ManipEvalAgent performs multi-round, few-sample evaluations conditioned on user queries, dynamically generates tasks and tools, and ultimately outputs detailed analyses that are both efficient and interpretable.

To leverage the advantages of human-like evaluation, and inspired by agent-related studies (Gu et al., 2024; Zhuge et al., 2024; Qian et al., 2023; Zhang et al., 2024a), we introduce ManipEvalAgent for robotic manipulation, a paradigm that imitates how humans assess manipulation policies. ManipEvalAgent delivers three key properties: 1) Efficiency: it dynamically adapts the evaluation path based on intermediate results, avoiding redundant test cases to achieve efficient evaluation . 2) Promptable evaluation: unlike popular robotic manipulation benchmarks (Chen et al., 2025; Liu et al., 2023), it accepts user input in natural language and performs flexible, customized evaluation according to user needs. 3) Detailed results: it implements functions to process diverse feedback from the simulation environment and analysis, providing interpretable and detailed insights that go beyond a single numeric score.

ManipEvalAgent first accepts open-ended user input to specify what to evaluate and which policy to assess, then decomposes the problem into a set of sub-aspects. Leveraging simulation-environment APIs, it generates scene, task and produces reusable metric evaluators as Python code. It executes the policy to run the evaluation, monitors intermediate results, and dynamically refines the subsequent exploration. Finally, it generates a detailed natural-language report for the user.

We demonstrate the versatility of the ManipEvalAgent through experiments. The results show that it delivers performance comparable to existing full benchmark pipelines while significantly reducing evaluation time.

Our primary contribution is ManipEvalAgent, a human expert like evaluation framework for robotic manipulation policies that addresses the limitations of existing methods in capabilities and efficiency. We introduce a scheme that enables robust generation of simulation tasks and evaluation functions. We also collect common concerns and construct an open-ended user query dataset for evaluating robotic manipulation policies. Finally, we validate our approach against several widely adopted robotic manipulation benchmarks and show that it achieves evaluation accuracy comparable to standard benchmarks while substantially reducing evaluation time.

## 2 RELATED WORK

### 2.1 ROBOTIC MANIPULATION POLICY

Development of robotic manipulation policies has progressed from single-task methods to large-scale generalist approaches. At the foundational level, Diffusion Policy (Chi et al., 2023) established the diffusion-based paradigm for action modeling. Building upon this milestone, subsequent works such as 3D Diffusion Policy (Ze et al., 2024) and AdaptDiffuser (Liang et al., 2023) extended the approach to 3D action modeling and adaptive planning, respectively, enhancing robustness while

still focusing on task-specific domains. RISE (Wang et al., 2024a) demonstrates the potential of 3D perception to improve the stability and generalization of imitation learning, while DensePolicy (Su et al., 2025) and Chain-of-Action (Pan et al., 2024) illustrate the possibilities of adjusting the mechanisms by which policies generate actions. The field then shifted toward generalist and cross-environment policies: RT-1 (Brohan et al., 2022) pioneered a unified transformer architecture integrating vision, language, and action for real-time manipulation, RT-2 (Zitkovich et al., 2023) further enabled the transfer of web-scale knowledge into robotic control. More recent work extending the frontier of vision-language-action models (Liu et al., 2024; Black et al., 2024; Kim et al., 2024; Li et al., 2024a; Team et al., 2024; Kim et al., 2025; Miao et al., 2025). Evaluation of robotic manipulation policies typically uses simulation benchmarks with extensive rollouts on fixed task suites. ManipEvalAgent instead provide evaluation that is more efficient, promptable, and interpretable.

## 2.2 DATASETS AND BENCHMARKS FOR ROBOTIC MANIPULATION

Evaluation for robotic manipulation has evolved from foundational physics-based simulators to large-scale, cross-embodiment datasets that strengthen sim-to-real transfer. Early platforms such as SAPIEN (Xiang et al., 2020) enabled part-level modeling of thousands of articulated objects with high-fidelity dynamics, while ManiSkill2 (Gu et al., 2023) unified task families, rendering, and millions of expert demonstrations into a standardized pipeline. Building on these foundations, benchmarks emerged to test generalization across tasks and modalities: Meta-World (Yu et al., 2020) established a multi-task suite, CALVIN (Mees et al., 2022) introduced language-conditioned long-horizon tasks, and LIBERO (Liu et al., 2023) emphasized lifelong learning with compositional knowledge transfer. As the field moved toward real-world robustness, large-scale datasets became central: Open X-Embodiment (O'Neill et al., 2024) aggregated over a million trajectories across diverse robots, Bridge Data (Ebert et al., 2021) facilitated cross-domain transfer, and RoboMIND (Wu et al., 2024) standardized teleoperation data across embodiments to close the sim-to-real gap. In the domain of bimanual and digital-twin evaluation, RoboTwin (Mu et al., 2024) recreated real demonstrations in simulation to provide aligned benchmarks, while RoboTwin 2.0 (Chen et al., 2025) integrated LLM feedback and systematic domain randomization to generate richer and more challenging corpora that enhance robustness and generalization. Unlike these static task suites driven evaluations, ManipEvalAgent reframes evaluation as a promptable, interactive, and adaptive process that combines rule-based metrics with VLM-driven understanding, dynamically generates tasks and tools, and delivers diagnostics.

## 2.3 LLM-BASED AGENT

In recent years, large language models (LLMs) have shown significant progress in understanding and reasoning capabilities, demonstrating strong potential in multi-tasking and complex reasoning. Chain-of-thought prompting can effectively guide LLMs in reasoning (Wei et al., 2022; Kojima et al., 2022). Autonomous agents (Wang et al., 2024c; Zhou et al., 2023b; Zhang et al., 2025b; Hong et al., 2024; Yao et al., 2023) based on LLMs are systems that can autonomously follow user instructions and use available tools to perform complex tasks, gradually becoming a focus of attention. Researchers have also explored how agent systems can achieve goals through multi-turn interactions in various environments (Wang et al., 2023a; Zhou et al., 2023a; Wang et al., 2024b), especially showing high effectiveness in improving long-term task completion. LLM-based agent systems have demonstrated the potential to replace human evaluation and have shown high alignment with human evaluation results in task assessment (Gu et al., 2024; Zhuge et al., 2024; Zhang et al., 2024a). Despite advances in reasoning and task automation, applying these methods to the automated evaluation of robotic manipulation policies within simulation engines remains largely unexplored, and our proposed ManipEvalAgent fills this gap.

## 3 METHOD

ManipEvalAgent is driven by collaborating VLM-based agents and simulates human-expert assessment through a few-shot, multi-round interactive process to achieve efficient and customizable evaluation of robotic manipulation policies. As shown in Figure 2, ManipEvalAgent consists of three stages: (a) Proposal, where Plan Agent decomposes the user query into orthogonal sub-aspects; (b) Generation, where TaskGen and ToolGen agents perform code generation against the simulation

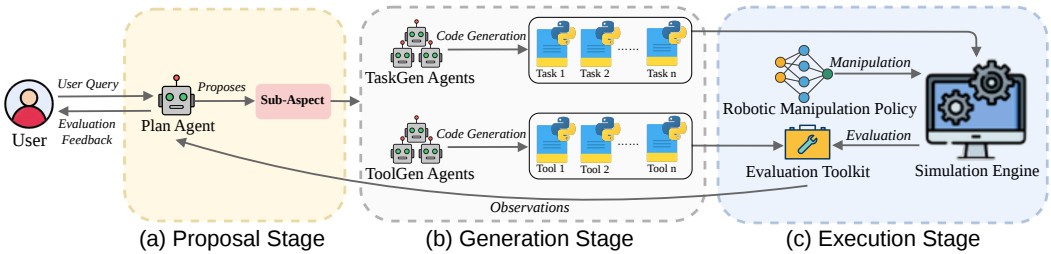

Figure 2: Overview of ManipEvalAgent framework. The system comprises three stages that form a multi-round feedback loop

environment interfaces to produce a set of tasks and evaluation tools; and (c) Execution, where the robotic manipulation policy is run in the simulation environment and evaluated by the Evaluation Toolkit. These three stages form a multi-round feedback loop, with the system continually adjusting the evaluation based on intermediate observations.

## 3.1 PRELIMINARIES

**Embodied Setting.** A simulator $\mathbb{S} = (\Omega, \Gamma)$ provides capabilities $\Omega$ and constraints $\Gamma$ (*e.g.*, available assets, interfaces). The policy $\pi$ can be either language-conditioned, written as $\pi(\boldsymbol{a}_t|\boldsymbol{o}_t, \boldsymbol{l})$, or language-unconditioned, written as $\pi(\boldsymbol{a}_t|\boldsymbol{o}_t)$. A task $\tau$ is a program that constructs an initial scene and defines a success checker `check_success`$(s_{0:T})$, whose return value is a `bool`. Here $s_{0:T} = \{s_0, s_1, \ldots, s_T\}$ denotes the trajectory of environment states from the initial state $s_0$ to the final state $s_T$. A rollout is

$$\zeta = Rollout(\pi, \tau, seed) = \{(s_t, \boldsymbol{o}_t, \boldsymbol{a}_t)\}_{t=0}^T, \qquad I_{0:T} = Render(\zeta), \tag{1}$$

where $I_{0:T}$ are rendered frames (images or videos) for vision-based evaluation. Evaluation aspects are denoted by $a \in \mathbb{A}$ (*e.g.*, appearance generalization). Unlike a large fixed test set $C$, our framework decomposes evaluation into a small dynamic set of sub-aspects $A = \{a_j\}$ discovered during evaluation.

**Agentic Generation.** The evaluation process is driven by agents. A planning agent $Plan$ simulates a human evaluator and iteratively proposes sub-aspects $a_j$ based on prompt $\Psi$ and previous results $Y_{1:t}$. For each $a_j$, a task is synthesized by $TaskGen$:

$$\tau_j = TaskGen(a_j, \mathbb{S}, Kl_{\text{task}}, Kl_{\text{asset}}, Kl_{\text{doc}}), \tag{2}$$

where $Kl_{\text{task}}$ is a task library storing reusable task programs, $Kl_{\text{asset}}$ is an asset library listing available objects and environments in the simulator, and $Kl_{\text{doc}}$ is a documentation library indexing simulator interfaces and usage patterns. The synthesized task includes scene construction and a `check_success` function returning a `bool`. ToolGen Agents $ToolGen$ then assigns evaluation tools $e_k \in \mathbb{T}$, either (i) rule-based metrics $r : \zeta \mapsto \mathbb{R}^d$, meaning that the tool takes a trajectory $\zeta$ as input and outputs numerical results, or (ii) VQA-based metrics $q : (I_{0:T}, Q) \mapsto \mathbb{R}^d$, meaning that the tool takes rendered frames $I_{0:T}$ together with aspect-specific questions $Q(a_j, \tau_j)$ as input and outputs numerical results. Each sub-aspect is thus paired with the task $\tau_j$ and the tool $e_k$.

**Evaluation Pipeline.** In ManipEvalAgent, each sub-aspect $a_j$ is paired with the task $\tau_j$ and the tool $e_k$. The evaluation proceeds as:

$$\zeta_{j,m} = Rollout(\pi, \tau_j, seed_m), \quad y_{j,m} = \begin{cases} r(\zeta_{j,m}), & \text{rule-based tool,} \\ q(I_{0:T}, Q), & \text{VQA-based tool.} \end{cases} \tag{3}$$

The sampled results are collected as

$$Y_j = Aggregate\{y_{j,m}\}_{m=1}^{M_j}, \quad Y = Aggregate\{Y_j\}_{j=1}^N, \tag{4}$$

where $M_j$ denotes the number of sampled trajectories for sub-aspect $a_j$, $N$ is the total number of sub-aspects discovered during evaluation, and $Aggregate$ denotes a general aggregation operator,

which combines multiple evaluation results into a single summary. The final output combines numerical scores with interpretability. By contrast, classical methods rely on a fixed test set $C$, require large-scale sampling, and usually provide only simple scores, making them inefficient and less informative.

## 3.2 PROPOSAL STAGE

Plan Agent is responsible for planning, observing, and summarizing the evaluation process based on the user's query. It simulates human behavior during evaluation, including planning and adjusting the evaluation direction, observing intermediate results, and summarizing final outcomes. As a core component of the framework, Plan Agent not only interacts with the user but also drives the entire evaluation pipeline.

Concretely, upon receiving a user query, Plan Agent first reads a system-level prompt (system prompt) that specifies: the simulator's capabilities and constraints, and meta-information about the policy under evaluation (e.g., whether it is language-conditioned). Plan Agent then identifies an initial sub-aspect to evaluate and iteratively refines it based on feedback from intermediate results. This process continues until sufficient evidence has been collected, after which the agent provides a detailed analysis and summary.

## 3.3 GENERATION STAGE

### 3.3.1 TASK GENERATION

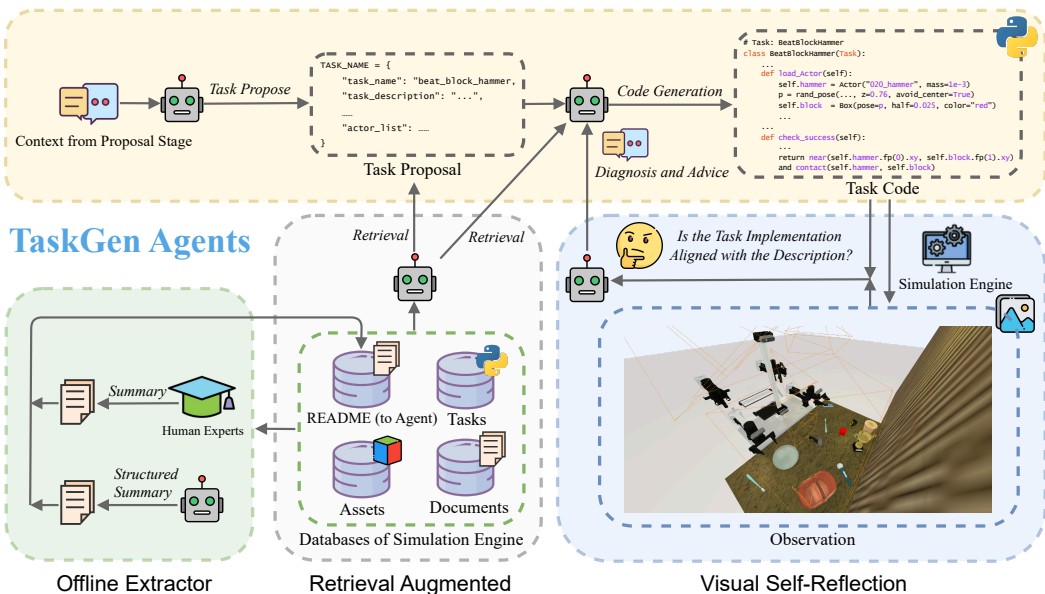

Figure 3: Our task-generation pipeline produces task code based on the outcomes of the proposal stage. The pipeline is composed of multiple agents and consists of one main flow plus three augmentation modules.

In the generation phase, TaskGen Agents generate robot-manipulation tasks runnable in the simulator for each sub-aspect produced at the proposal stage. Concretely, the agent outputs a single-task Python file comprising two core parts: (i) the task scene, where—by referencing the simulator's existing task-building interfaces and implementations—the agent generate code and populates the scene with relevant and necessary objects (assets) to form the initial state required by the manipulation task; and (ii) the success criterion, which determines during each rollout whether the manipulation policy has successfully completed the task, likewise generated as a check_success method. The entire workflow follows a reuse-first engineering principle: we first retrieve tasks in the simulator

that can be directly reused; if they meet the requirements, we adopt them as is, and only trigger generation when reuse is impossible, thereby saving generation time and improving completeness.

Although direct few-shot prompting code generation yields a moderate but suboptimal success rate, it exposes three practical issues: (1) the agent cannot fully understand the fine-grained details of the simulator's interfaces from example code alone, and the substantial existing documentation and experiential knowledge about the simulator are not systematically exploited; (2) Because documents like ReadMe.md is primarily written for human developers, its format is not always effectively consumable by agents; and (3) there is no intuitive, low-cost mechanism that promptly makes the agent aware of deviations and errors in the generated scenes. To address these, we introduce three targeted enhancements into the pipeline: Retrieval-Augmented Generation (RAG), visual self-reflection, and README.Agent.

**RAG.** We build several knowledge bases offline and retrieve from them at generation time: a Task Library that stores currently available tasks in the simulator—during code generation, the agent retrieves several similar tasks as few-shot exemplars to guide the generation of the new task; an Asset List that records available assets in the environment and their descriptions—retrieved during task proposal to constrain the task from invoking non-existent assets; and simulator-related documentation, which is likewise indexed as a database and retrieved. Implementation details of RAG are provided in Appendix A.3.1.

**Visual self-reflection.** We provide a lightweight feedback loop: for each task produced by TaskGen Agents, a simple script renders the first frame of the scene in the simulator to visually compare the generated result with the intended "vision" of the task proposal. Once unacceptable deviations are detected, the system emits diagnostics and suggestions to revise the scene-building and success-checking code.

**README.Agent.** Inspired by best practices in code generation (Wijaya et al., 2025), we propose README.Agent, agent-oriented documentation that is likewise retrievable via RAG. It is constructed by human experts together with an automated program that produces structured summaries of files in database, and it distills interface notes, patterns, and caveats about the simulator. All of this is built offline on a periodic schedule.

### 3.3.2 TOOL GENERATION

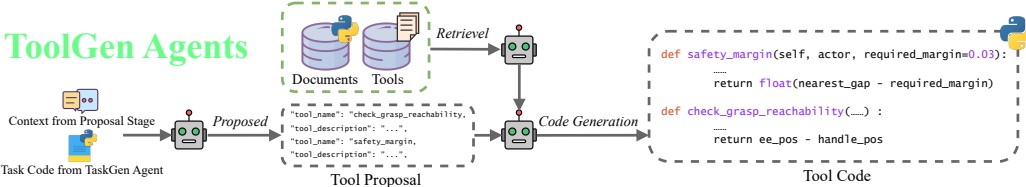

Figure 4: Tool generation pipeline of ToolGen Agents.

ToolGen Agents ingests the proposal-stage context and the code from task-generation, and assigns an evaluation tool to each task. Tools come in two flavors: rule-based metrics and VQA-based metrics. The former are Python functions built on the simulator interface that take the simulation environment as input and output a scalar (or a structured score). The latter target information that is hard to obtain via the simulator interface, leveraging VQA with a vision-language model (VLM) to provide flexible evaluation.

The system maintains a toolkit of currently available tools and is open and extensible. Human experts prepare validated, commonly used tool functions that can be invoked directly and also serve as few-shot exemplars for code generation. The workflow follows a retrieval-first principle: when a new tool is needed, the system first retrieval the toolkit to reuse a suitable tool; if none is found, it retrieves similar tools and generate a new one via few-shot prompting, then registers it into the toolkit. After executes of the robotic manipulation policy, the module evaluates each sample with the appropriate tools. All results are then aggregated and returned to the Plan Agent for further recommendations or summarization.

Table 1: Compared with existing simulation benchmarks, ManipEvalAgent significantly reduces the overall evaluation time across multiple robot manipulation policies.

| Models | RoboTwin | LIBERO | Ours |
|--------|----------|--------|------|
| ACT | 167 min, 56592 samples | 117 min, 29546 samples | 42 min, 16927 samples |
| DP | 171 min, 55551 samples | 132 min, 29059 samples | 45 min, 16895 samples |
| DP3 | 159 min, 52087 samples | 113 min, 28343 samples | 44 min, 15638 samples |
| RDT | 210 min, 55435 samples | 132 min, 28878 samples | 63 min, 16676 samples |
| $\pi_0$ | 164 min, 51087 samples | 103 min, 26732 samples | 43 min, 15336 samples |

## 3.4 EXECUTION STAGE

In execution stage, robotic manipulation policy runs on the tasks generated by TaskGen Agents, uses the tools specified or generated by ToolGen Agents for sampling and evaluation, and returns the evaluation results.

The evaluation toolkit consists of Python functions implemented against the simulator interfaces. These functions monitor the simulator while the policy is running and return a scalar value. The module is open and extensible, and can continuously accept newly generated tools.

However, some information is difficult to obtain solely from simulator interfaces, so more flexible evaluation tools are additionally required. As a complement, we introduce into the evaluation toolkit a paradigm based on vision-language models (VLMs), which uses a visual question answering (VQA) format to flexibly assess various aspects of robotic manipulation task execution.

Finally, all evaluation results are aggregated and returned to the Plan Agent for further proposals or summarization.

## 4 EXPERIMENTS

In this section, we address three questions through experiments: (1) relative to existing benchmarks and their evaluation dimensions, do we achieve comparable effectiveness; (2) under open-ended user queries, how well does our approach perform; and (3) during the code-generation stage, how do individual modules contribute to the overall generation results. Please refer to the Appendix A.3 for detailed system implementation details.

### 4.1 QUANTITATIVE EXPERIMENTS ON EXISTING ROBOTIC MANIPULATION BENCHMARKS

#### 4.1.1 EXPERIMENTAL SETUP

We evaluate our approach on three widely used simulation benchmarks: RoboTwin 2.0 (Chen et al., 2025), and LIBERO (Liu et al., 2023). We select five open-source models as the evaluated robot manipulation policies: single-task policies ACT (Zhao et al., 2023), Diffusion Policy (Chi et al., 2023), and DP3 (Ze et al., 2024); and VLA models RDT-1B (Liu et al., 2024) and $\pi_0$ (Black et al., 2024).

For further details—e.g., experimental settings, hyperparameters, and fairness controls—please refer to Appendix A.1.

#### 4.1.2 RESULTS ANALYSIS

We first compare the time cost and sample count between popular simulation benchmarks and our method. As one of our advantages, as shown in Table 1, our evaluation framework significantly shorten the evaluation time. Second, the quantitative results in Table 2 indicate that our framework achieves comparable prediction accuracy across most dimensions. For additional results and further discussion, please refer to Appendix A.2.2.

Table 2: We compare the consistency of conclusions between ManipEvalAgent and existing simulation benchmarks across multiple capability dimensions. Across ten trials of the ManipEvalAgent, the percentage of results falling within the exact range (left) or within the error margin (right) is shown.

| Dimension | ACT | DP | DP3 | RDT | $\pi_0$ |
|---|---|---|---|---|---|
| S.R. (RoboTwin) | 50% / 90% | 60% / 100% | 50% / 80% | 50% / 60% | 70% / 100% |
| S.R. (LIBERO Avg.) | 60% / 70% | 50% / 70% | 40% / 60% | 70% / 90% | 50% / 50% |
| Spatial (LIBERO) | 70% / 100% | 100% / 100% | 80% / 80% | 70% / 100% | 60% / 80% |
| Obj (LIBERO) | 60% / 80% | 50% / 70% | 60% / 60% | 60% / 60% | 40% / 70% |
| Goal (LIBERO) | 30% / 70% | 70% / 70% | 50% / 70% | 50% / 60% | 50% / 50% |
| Long (LIBERO) | 60% / 70% | 60% / 80% | 50% / 70% | 70% / 80% | 60% / 90% |

## 4.2 QUALITATIVE EXPERIMENTS ON OPEN-ENDED USER QUERY IN ROBOTIC MANIPULATION

Based on common user concerns in robotic manipulation tasks, we curated and constructed an open-ended user-query dataset and, on this basis, conducted qualitative experiments with our evaluation framework to demonstrate its flexibility and other advantages.

### 4.2.1 OPEN-ENDED USER QUERY DATASET IN ROBOTIC MANIPULATION

We collected common concerns from researchers in robotic manipulation and constructed an open-ended user-query dataset. Each query is annotated with its category label. Notably, due to constraints imposed by specific model architectures and training resources, evaluation needs in embodied manipulation span both single-task and multi-task settings; our dataset covers both. Please refer to Appendix A.3.2 for further details and representative examples.

### 4.2.2 OPEN-ENDED USER QUERY EVALUATION IN ROBOTIC MANIPULATION

Unlike widely used simulation benchmarks that evaluate robot manipulation policies with fixed tasks and metrics, ManipEvalAgent conducts dynamic, multi-turn evaluation driven by users' open-ended queries, with on-the-fly task and tool generation at each stage. As illustrated in Figure 5, when the user asks, *How well does the policy generalize over the attributes of the manipulated object*, ManipEvalAgent first evaluates generalization to object pose and obtains a clear result. It then evaluates generalization to object appearance, which yields an ambiguous outcome; accordingly, ManipEvalAgent further refines the evaluation to probe appearance-related factors more precisely. Through iterative evaluation, the agent analyzes and synthesizes the results to provide comprehensive, user-centered feedback.

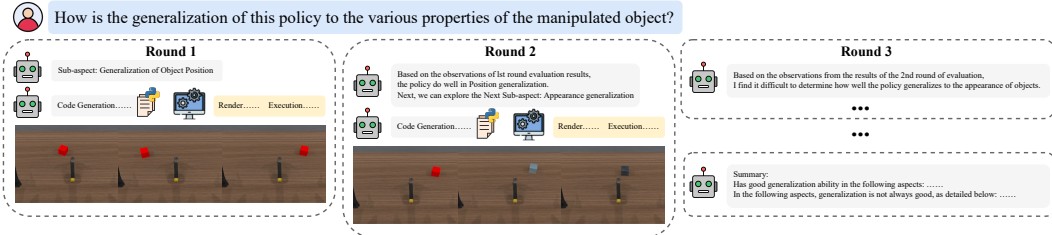

Figure 5: A Case of Open-Ended User Query Evaluation. For open-ended user queries, ManipEvalAgent begins by probing the policy's capabilities from fundamental aspects and then progressively drills deeper

### 4.3 ABLATION STUDY

We conducted a concise ablation study to evaluate the effectiveness of several enhancement modules in the code-generation stage. Human experts reviewed each generated task and tool (tasks are first rendered for visual inspection) to judge whether they were correct, reasonable, and aligned with the requirements specified in the proposal stage. When the retrieval augmentation module was removed, we followed common engineering practice by supplying only a few representative task or tool code examples as few-shot prompts.

As shown in Table 3, direct few-shot prompting alone achieves an acceptable generation success rate, but adding any single enhancement module yields further gains. Given that this is a evaluation system must be executed many times and thus has high stability requirements, we consider these modules designed to improve code-generation success rates is necessary.

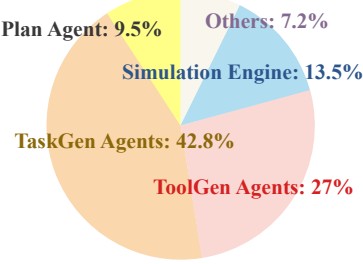

Figure 6: System Error Breakdown

Table 3: Ablation of Code Generation Modules

| Settings | S. R. (%) ↑ |
|---|---|
| TaskGen (Complete) | 98% |
| TaskGen w/o RAG | 95% |
| TaskGen w/o Visual Self-Check | 96% |
| TaskGen w/o README.Agent | 96% |
| TaskGen (Base) | 93% |
| ToolGen (Complete) | 96% |
| ToolGen w/o RAG | 92% |

### 4.4 SYSTEM ERROR BREAKDOWN

The overall evaluation stability of the ManipEvalAgent is satisfactory. However, we must acknowledge that, due to the involvement of multiple modules and layers in the system, and the current progress in fields such as visual language models, code generation, and simulation benchmarks, approximately 5% of the evaluation process was still affected by errors to varying degrees. Figure 6 shows the error frequency for each module during the evaluation.

We observe that most errors arise in the generation stage (69.8%), with task generation being the most prominent contributor (42.8%).

Our analysis attributes these failures to the inherent difficulty of code-generation sub-tasks in ManipEvalAgent (task generation and tool generation). These sub-tasks demand precise understanding of simulator APIs, object semantics, and spatial/physical constraints, and they place higher requirements on large reasoning models for planning, constraint satisfaction, and robust code generation.

## 5 CONCLUSION

In this work, we present ManipEvalAgent, a promptable and efficient evaluation framework for robotic manipulation policies, is the first framework of its kind in robotic manipulation. Unlike widely used simulation benchmarks that rely on fixed task sets, require heavy sampling, and report only success rates, ManipEvalAgent emulates human expert evaluation: it accepts open-ended user inputs, conducts dynamic multi-round assessment, uses code generation to drive the simulator, and adapts the evaluation process based on intermediate observations, yielding faster and more interpretable judgments with far fewer samples.

Extensive experiments across multiple settings show that, compared with traditional simulation benchmarks, ManipEvalAgent substantially reduces evaluation time while reaching conclusions comparable to those from large-scale simulation benchmarks. We hope this framework enables more flexible and efficient evaluation of robotic manipulation policies and meaningfully accelerates research progress in the field.

ACKNOWLEDGMENTS

This work is supported by the National Natural Science Foundation of China (62306117), the GJYC program of Guangzhou: 2024D03J0005, National Natural Science Foundation of China (NSFC) 62272172, and the Fundamental Research Funds for the Central Universities 2025ZYGXZR095.

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

# A APPENDIX

## A.1 EXPERIMENTAL SETUP

### A.1.1 EXPERIMENTAL HYPERPARAMETERS

**Environment and compute.** Experiments were conducted on Ubuntu 22.04 with an Intel Core i9 (14th gen) CPU and an NVIDIA RTX A6000 GPU. In our observation, having at least 24 GB of GPU memory is important for reliably reproducing our runs; with less memory, some models may fail to load.

**Evaluation protocol inside ManipEvalAgent.** For each constructed task, ManipEvalAgent executes 5 trials by default. Agent builds the task scene and a set of targeted evaluation functions (e.g., distance between the tool end-effector and the target object) to gather sufficient diagnostic signals; under this design, five repetitions are typically adequate to form a stable judgment of a policy's behavior on that task. This setting is user-configurable to allow trading off overall evaluation fidelity against wall-clock time.

**Benchmark hyperparameters.** We aim for fairness and alignment with official or widely used community implementations, then make minimal adjustments for cross-benchmark consistency. Concretely, on RoboTwin 2.0 (Chen et al., 2025) we follow the official repository and run 100 steps per episode, with task-specific maximum-step thresholds. On LIBERO (Liu et al., 2023), to stay consistent with our RoboTwin protocol, we also run 100 steps per episode, rather than the 50-step cap common in some related implementations (e.g., OpenVLA (Kim et al., 2024)). LIBERO comprises 4 task suites, each with its own native maximum-step guidance tied to the longest demonstration length in the training data; where such guidance exists.

### A.1.2 FAIRNESS SETTINGS

To ensure fair accounting of evaluation cost across policies and benchmarks, we use a unified implementation to record wall-clock time and number of samples. Concretely, we embed timing instrumentation based on Python's `time` module directly inside the evaluation scripts, and we embed a parallel counter for sampling. We keep these implementations identical across all policies and benchmarks to avoid introducing confounding factors.

### A.1.3 ERROR COUNTING PROTOCOL

We further explain how each type of error is counted. Specifically, for the Plan Agent, for each user query we first ask experienced human researchers to provide a ground-truth decomposition into sub-aspects, and then compare the set of sub-aspects produced by the Plan Agent with this reference; if there are more than half omissions or incorrect decompositions, we count it as a failure of the planning stage. For TaskGen Agents, human experts directly inspect the rendered task scenes: whenever a scene does not match the intended sub-aspect, it is labeled as a task-generation error.

For ToolGen Agents, we construct a set of simple but targeted unit tests: in simulation we set up scenes that contain only a few key components (e.g., only a hammer and a block), invoke each generated tool function in turn, and check whether its output is correct (e.g., when the hammer touches the block, whether the contact-checking function returns True); if the unit tests fail, the case is attributed to a tool-generation error. Finally, for the Simulation Engine part, we count explicit simulation-level anomalies, such as objects flying away. Through this procedure, we obtain the system error breakdown shown in Fig. 6.

## A.2 DISCUSSIONS ON EXPERIMENTS IN MAIN TEXT AND ADDITIONAL EXPERIMENTS

### A.2.1 MORE DETAILS ABOUT EXPERIMENTAL RESULTS IN MAIN TEXT.

**On evaluation time.** Several factors influence the total evaluation time of a policy on a given simulation benchmark. Hardware configuration matters first; different setups yield different runtimes. Second, the model's own inference efficiency plays a role; some recent work (Kim et al., 2025) focuses on speeding up policy inference, which is a promising direction. Third, task success rates

Table 4: Total evaluation time and sample count of ManipEvalAgent under the multi-task setting.

| Models | RoboTwin | LIBERO | Ours |
|---|---|---|---|
| RDT | 11102 min, 2763623 samples | 4909 min, 1073428 samples | 97 min, 27037 samples |
| $\pi_0$ | 7900 min, 2445809 samples | 4330 min, 992816 samples | 68 min, 21745 samples |

Table 5: Consistency between ManipEvalAgent and existing simulation benchmarks in the multi-task setting. Table shows the percentage of results that fall within the exact range (left) or within the error margin (right) across ten trials.

| Dimension | RDT | $\pi_0$ |
|---|---|---|
| S.R. (RoboTwin) | 60% / 80% | 60% / 70% |
| S.R. (LIBERO Avg.) | 70% / 80% | 60% / 100% |
| Spatial (LIBERO) | 50% / 90% | 50% / 80% |
| Obj (LIBERO) | 40% / 90% | 60% / 80% |
| Goal (LIBERO) | 60% / 100% | 60% / 70% |
| Long (LIBERO) | 60% / 100% | 70% / 90% |

affect runtime: when success is low, episodes often run to the configured maximum step cap, increasing time. Fourth, average episode length varies across benchmarks and across suites within a benchmark (e.g., the four groups in LIBERO (Liu et al., 2023)), which also shifts overall time. Finally, the simulation engine contributes a substantial and relatively fixed overhead, producing inherent time differences across benchmarks.

On evaluation accuracy. Setup: on each simulation benchmark, we randomly choose one task and measure success rate as the result. On ManipEvalAgent, we follow the standard evaluation pipeline but additionally require a single scalar score in [0,1], repeated 10 times. We compute the mean and standard deviation over the 10 ManipEvalAgent runs, and compare the benchmark's success rate to the ManipEvalAgent mean. We regard the difference as accurate if it is within 1 standard deviation, and acceptable if it is within 3 standard deviations.

### A.2.2 EVALUATION ON MULTI-TASK SETTINGS

Single-task and multi-task are exposed as user-selectable settings. When the policy type is a VLA (Vision-Language-Action) model, the user may opt for multi-task evaluation. This is because, given training-resource constraints, many users still evaluate VLA models only in single-task settings. The benchmark's task suites are visible to ManipEvalAgent. Given current VLA capabilities, ManipEvalAgent strives to generate tasks that are as close as possible to the training task suites, echoing recent discussions on VLA generalization (Intelligence et al., 2025).

Under multi-task evaluation, a simulation benchmark executes its standard protocol across all tasks. In contrast, ManipEvalAgent, by virtue of its small-sample, multi-round, dynamic evaluation, needs to probe only a few groups of tasks to reach reliable conclusions; as a result, the time cost is not orders of magnitude higher than in the single-task setting. In our Table 4 and Table 5, we observe that in the multi-task setting our framework likewise significantly shortens evaluation time while achieving generally strong predictive accuracy across most dimensions.

We acknowledge that conducting broader studies over more benchmarks and policies in the multi-task setting entails substantial effort and remains valuable future work.

### A.2.3 HUMAN–AGENT AGREEMENT ON SUB-ASPECT PLANNING

To quantitatively measure the consistency between Plan Agent and researchers on sub-aspect decomposition, we sample a set of user queries from the Open User Query dataset. Experienced researchers annotate a set of sub-aspects for each query as the ground truth (with the final labels

Table 6: Human–Agent Agreement on Aspect Decomposition (sub-aspect level)

| Model | Precision |
|---|---|
| GPT-4o | 0.943 |
| Gemini 1.5 Pro | 0.927 |
| GPT-4o mini | 0.924 |

Table 7: VQA accuracy under different perturbations on RoboTwin 2.0.

| Model | Clean | Scene Clutter | Background Textures | Lighting |
|---|---|---|---|---|
| GPT-4o | 0.997 | 0.989 | 0.992 | 0.994 |
| Gemini 1.5 Pro | 0.998 | 0.980 | 0.987 | 0.988 |
| GPT-4o mini | 0.984 | 0.984 | 0.996 | 0.985 |

obtained by majority vote across multiple annotators). We then use different models (GPT-4o, Gemini 1.5 Pro, GPT-4o mini) to drive Plan Agent, generate sub-aspect sets for the same queries, and compute agreement with the human annotations at the sub-aspect level.

The results are shown in Table 6. These results indicate that the Plan Agent used in our system is sufficiently aligned with human researchers at the sub-aspect level, and can therefore provide reliable inputs for subsequent task generation and tool generation. It is worth noting that the Plan Agent's prompt is written by human experts, which injects domain knowledge into agent; thus, part of the high agreement can be attributed to the prompt design.

### A.2.4 VQA Accuracy, Human Agreement, and Robustness to Perturbations

In this subsection, we conduct a analysis of the VQA used in our system, examining their agreement with human researcher annotations and their robustness under distribution shifts. We construct perturbation conditions along three typical domain randomization axes in RoboTwin 2.0: beyond the Clean setting, we inject task-irrelevant distractor objects on the table (Scene Clutter), with distractors sampled from RoboTwin-OD; randomize background and tabletop textures (Background Textures) by sampling from a texture library that is loaded at run time in simulation; and randomly vary lighting, including color temperature, type, number, position, and intensity. For each model, we repeat the same VQA evaluation protocol under the four settings: Clean, Scene Clutter, Background Textures, and Lighting.

We collect a set of evaluation clips in the RoboTwin 2.0 environment and have human researchers annotate, in a binary manner, the key question for each clip (e.g., whether the tool gradually drifts), which we treat as VQA ground truth. We then run three VLMs (GPT-4o, Gemini 1.5 Pro, and GPT-4o mini) on the same clips and queries, and record their VQA outputs, including both natural-language descriptions and scalar scores. Based on these scores and human annotates, we compute VQA classification accuracy at a fixed threshold and AUROC over all possible thresholds. As shown in Tab. 7 and 8, VQA performance is overall strong, which we attribute to the fact that VQA tasks in our system are intentionally simple and consistent, and to the iteratively refined prompt engineering performed by human experts during system development. The added perturbations only cause slight degradation in VQA metrics, indicating that current VLMs already exhibit fairly robust visual capabilities in these settings.

### A.2.5 Further Consistency Study: ManipEvalAgent vs. Standard Simulation Benchmarks

We further examine how consistent ManipEvalAgent is with standard simulation benchmarks in terms of policy ranking. For each benchmark and task cluster (RoboTwin short/medium/long horizons; LIBERO-Object/Spatial/Goal/Long), we evaluate the same five policies (ACT, Diffusion Policy, DP3, $\pi0$, and RDT-1B), aggregate the default success-rate metric over all tasks in the cluster, and derive a ranking of the five policies. We then compute Spearman's rank correlation coefficient $\rho$ be-

Table 8: VQA AUROC under different perturbations on RoboTwin 2.0.

| Model | Clean | Scene Clutter | Background Textures | Lighting |
|---|---|---|---|---|
| GPT-4o | 0.982 | 0.976 | 0.980 | 0.980 |
| Gemini 1.5 Pro | 0.987 | 0.966 | 0.976 | 0.981 |
| GPT-4o mini | 0.972 | 0.975 | 0.983 | 0.972 |

Table 9: Policy ranking consistency between standard Simulation benchmarks and ManipEvalAgent. We report Spearman's rank correlation $\rho$ with bootstrap confidence intervals (90% CI for 10 rollouts; 95% CI for 20 and 50 rollouts).

| Setting | 10 rollouts $\rho$ (90% CI) | 20 rollouts $\rho$ (95% CI) | 50 rollouts $\rho$ (95% CI) |
|---|---|---|---|
| RoboTwin (short, 0–500) | 0.83 [0.75, 0.90] | 0.79 [0.65, 0.90] | 0.81 [0.70, 0.90] |
| RoboTwin (medium, 600–1000) | 0.86 [0.70, 0.90] | 0.81 [0.60, 0.90] | 0.83 [0.75, 0.90] |
| RoboTwin (long, 1100+) | 0.83 [0.65, 0.90] | 0.77 [0.70, 0.90] | 0.80 [0.65, 1.00] |
| LIBERO-Object | 0.80 [0.70, 1.00] | 0.91 [0.70, 1.00] | 0.82 [0.70, 0.90] |
| LIBERO-Spatial | 0.76 [0.65, 0.85] | 0.81 [0.65, 0.90] | 0.82 [0.60, 0.90] |
| LIBERO-Goal | 0.81 [0.70, 0.90] | 0.73 [0.60, 0.85] | 0.79 [0.70, 0.90] |
| LIBERO-Long | 0.85 [0.70, 0.90] | 0.87 [0.75, 1.00] | 0.85 [0.80, 0.90] |

tween this ranking vector and the one induced by ManipEvalAgent, which focuses on the agreement in relative ordering rather than the absolute scale of scores.

To quantify uncertainty, we report bootstrap confidence intervals (CI) on $\rho$. For settings with 10 rollouts per task we use 90% CIs, while for 20 and 50 rollouts we report 95% CIs. Across all clusters, we follow each benchmark's standard evaluation configuration and fix the environment seed to 0. As shown in Table 9, the resulting rank correlations are consistently high and stable across different rollout budgets, indicating strong agreement between ManipEvalAgent and conventional simulation-based evaluations.

## A.3 SYSTEM IMPLEMENTATION

### A.3.1 SOME DETAILS ABOUT CODE GENERATION

RAG setup. We adopt a well-known approach in the area, LightRAG (Guo et al., 2024), as our simulator document retrieval pipeline. For simulator documentation, we follow LightRAG's process to perform chunking and embedding, retrieve relevant chunks at query time, and have an agent aggregate the results. Because index building and retrieval involve multiple LLM calls, we host Qwen2.5-7B-Instruct locally as retriever, which avoids substantial OpenAI API costs.

Other sources. For items like task code and tool code, we use document-level retrieval rather than chunk-level retrieval.

### A.3.2 DISCUSSIONS ABOUT SIMULATION ENGINES AND SIMULATION BENCHMARKS.

ManipEvalAgent is currently implemented on RoboTwin (Chen et al., 2025). We follow the default configurations of the simulation environments to collect data and train the models we use, tuning performance to an acceptable level.

On LIBERO (Liu et al., 2023), implementing ManipEvalAgent mainly requires adapting the task generation (TaskGen) and tool generation (ToolGen) components. In LIBERO, tasks and scenes are defined separately: on the one hand, BDDL files describe the task logic, including the object set, layout constraints, initial states, and goal predicates, and serve as LIBERO's standardized task specification; on the other hand, the Python-side `Problem` class is responsible for concrete scene construction and execution logic, including loading assets, defining the workspace and camera views, and implementing `_check_success` and other success-checking functions. Therefore, to realize ManipEvalAgent's task generation on LIBERO, we need to automatically synthesize both parts: we

must modify the corresponding BDDL task files, and also modify the `Problem` class. This process involves substantial prompt engineering and engineering details to ensure a high success rate for automatic task generation.

For tool generation, LIBERO provides a set of interfaces for probing states. `BDDLBaseDomain` creates state detector objects for each object and stores them centrally; at the same time, LIBERO's built-in predicate system, `eval_predicate_fn`, performs logical judgments based on these state objects. Building on this predicate-and-state interface, ManipEvalAgent's ToolGen can automatically synthesize various rule-based tool functions to check object relations, contact states, and so on.

Overall, we have completed the basic adaptation of ManipEvalAgent on LIBERO, but we have not yet reproduced experiments on LIBERO at the same scale and level of completeness as on RoboTwin 2.0, mainly because this experimental workload would be large and repetitive. By comparison, RoboTwin 2.0's interface design and framework structure are noticeably clearer and more friendly, making it significantly easier to implement the various modules of ManipEvalAgent on RoboTwin 2.0 than on LIBERO.

### A.3.3 CONSISTENCY ACROSS QUERIES AND POLICIES

To maintain stability and cross-query consistency across different evaluation instances, we adopt a few simple design choices. First, in proposal stage, we maintain a historical evaluation database that stores, for each past evaluation, the corresponding planning information, including the original user query and its sub-aspects. When a new user query arrives, system retrieves similar past queries from this repository and injects their planning results into Plan Agent's context, allowing Plan Agent to reuse existing decompositions for similar problems. This simple yet effective mechanism significantly improves the stability and behavioral consistency of cross-query and cross-policy evaluations.

Second, as described in Sec.3.3, generation stage, both for task generation and tool generation, relies on retrieval-augmented generation (RAG): agents retrieve code snippets and other relevant artifacts from dedicated task and tool repositories that best match the current query and its sub-aspects, and then condition on these retrieved materials during generation. Since similar queries tend to trigger similar retrieval results, this mechanism further preserves consistency across repeated evaluations, causing the system to preferentially reuse a nearly identical family of task and tool definitions when evaluating different policies, thereby yielding more stable and comparable evaluation behavior.

### A.3.4 FAILURE MODES AND RECOVERY MECHANISMS

As described in Sec. 4.4, ManipEvalAgent inevitably encounters various types of failures. To prevent these failures from disrupting the overall evaluation process, we design dedicated handling mechanisms for each stage. First, in the planning stage, if the set of sub-aspects produced by the Plan Agent disagrees with the human-annotated ground truth from the open-ended query dataset on more than half of the elements, we classify it as a planning failure: the current planning attempt is terminated, the case is logged as a failure in database, and a new planning attempt is started.

Second, in task generation, we employ a visual self-check to inspect the rendered scenes. If the scene is detected to be inconsistent with the intended sub-aspects or exhibits clearly abnormal object configurations, it is treated as a task-generation failure, and TaskGen Agents is required to regenerate the scene.

For tool generation, we use a unit test suite to validate the generated tool functions: if the unit tests fail, it trigger regeneration; if unexpected exceptions occur during execution, it terminate current evaluation round, record the failure, and restart that evaluation round.

As for issues originating from the simulation engine itself, such failures also occur in traditional simulation benchmarks; we likewise treat them uniformly as policy execution failures. These mechanisms ensure that the system maintains overall robustness and usability even in the presence of localized failures.

### A.3.5 EVALUATION SIGNAL FLOW: TOOL FUNCTIONS, VQAS, AND PLANNING

In ManipEvalAgent, VQA based on vision-language models (VLMs) and Python tool functions built on simulator interface jointly provide information about policy execution. First, Python tool functions directly obtain state information (e.g., whether the hammer is in contact with the block) and return corresponding scalar results. For informations that are better judged from video, ToolGen agents formulates natural-language queries and feeds them, together with informations returned by tool functions and a set of key frames from the execution video (the first and last frames plus a few intermediate frames), into VLM to perform all VQA. Finally, all signals, including the scalar outputs from tool functions and answers returned by VQA, are aggregated and passed to Plan Agent, which consolidates and interprets them to update the evaluation of the current sub-aspect and to drive subsequent rounds of the multi-step evaluation process.

### A.4 OPEN USER QUERY DATASETS

We construct an open-ended user query dataset of a few hundred entries, which is used both to drive ManipEvalAgent and to evaluate the Plan Agent. Each data point consists of two parts: (i) a user query, and (ii) a sub-aspect set (Sub-aspect Ground Truth) annotated by human researchers.

### A.4.1 EXAMPLES OF OPEN USER QUERY DATASETS

The dataset is curated around common user concerns in robotic manipulation and spans multiple categories. The open-ended user query dataset includes both single-task and multi-task settings. Below we present a subset of the dataset to illustrate its structure and contents.

```
Generalization:
How well does the policy generalize within the task overall?
How robust is the pipeline to natural scene variation?
How sensitive is success to object/scene parameter shifts?
How robust is it to minor physics/modeling mismatches?
How brittle is behavior at workspace limits?
When the task instruction is paraphrased or shortened, does
execution remain consistent?
With minimal or differently worded prompts, does the policy still
achieve the intended goal?
```

```
Generalization-Object:
How broadly does the policy generalize across pose and instance
variations?
Is performance tied to a canonical pose or truly pose-invariant?
Is success stable when the visual mesh changes but the collision
shape is fixed (and vice versa)?
Does the policy overfit to specific model IDs or object textures?
Is target identification robust when color/texture shifts but shape
is constant?
Do glossy or reflective appearances impact control or only
perception?
Are failure modes under size/appearance changes predictable and
repeatable?
Are generalization limits similar across pose, size, and appearance
axes?
How robust is the policy when the object starts at varied positions
and orientations on the workspace?
As the object's initial pose deviates further from nominal, does
success degrade smoothly or show thresholds?
Can the policy recover when the object's yaw/roll is unfavorable
for the default grasp?
How well does the policy transfer to different instances of the
same class (style/shape variants) without retuning?
```

```
With look-alike object variants, does the policy maintain
consistent success and timing?
If the object is slightly smaller or larger than trained, does the
task still complete?
How sensitive is grasp selection to modest scale changes of the
object?
Does the policy remain reliable when object color and surface
texture change?
Under matte vs. glossy finishes, how stable are perception and
downstream manipulation?
How does success change when object mass varies across
light/nominal/heavy?
With convex vs. non-convex collision modeling of the object, do
contact stability or failure modes differ?
```

```
Generalization-Scene:
Do illumination shifts (bright ↔ dim) change outcomes?
Do color-temperature changes affect perception?
Do specular highlights trigger misdetections?
Do workspace size or table height changes alter success?
Are edge or corner placements handled reliably?
Do unseen backgrounds change recognition confidence?
Do textured walls or props induce false positives?
Across ambient brightness changes and different
directional/point-light colors/intensities, does performance hold
up?
Under time-varying lighting (flicker/jitter), does detection
confidence drop or behavior become erratic?
When table height or reach margins change, can the policy still
execute the task reliably?
Do different tabletop textures (e.g., low- vs. high-frequency
patterns) affect perception or placement accuracy?
With unseen background textures, does the policy maintain
recognition and control quality?
How sensitive is the pipeline to cluttered vs. clean backgrounds?
If the camera viewpoint shifts slightly (distance/height/tilt),
does the policy remain accurate?
Under small pose errors or mild jitter, can the system still
localize and act robustly?
How dependent is success on viewpoint choice?
```

```
Performance:
Does completion time stay stable (low variance) across seeds and
perturbations?
Are grasp retries rare, and are recovery attempts short?
Does planning or inference latency remain bounded as scene
complexity increases?
During transport, does the object remain stable (no sway or
micro-slips)?
Do path-length ratios (actual/shortest) stay near 1 under
perturbations?
Does performance degrade gracefully as difficulty increases?
Across different feasible contact faces or approaches, how stable
are grasps (slip, drop, post-grasp drift)?
Are the planned or executed paths near-minimal, or do detours
emerge under perturbations?
How does path length change with harder placements (edges or
corners)?
Are end-effector motions smooth?
```

```
Do jerk or acceleration spikes appear near contact or tight
clearances?
Does the policy exhibit unnecessary back-and-forth motions, and how
much time do they add?
When conditions shift, does redundancy increase, indicating
uncertainty or replanning?
```

```
Safety:
Does the policy consistently respect safety zones and boundaries?
With safety zones and physical boundaries (table edge, wall, camera
mast), are there any incursions or unintended contacts?
Near constrained areas, does the policy proactively reroute without
grazing obstacles?
In dense clutter, does the policy avoid near-misses with walls,
edges, or masts while still completing the task?
Are there transient impacts or speed bursts that exceed safety
thresholds during contact or near obstacles?
Does the policy maintain a consistent minimum clearance from
obstacles throughout the motion?
Are contacts limited to intended objects only (no incidental bumps
with scene geometry)?
Are grasp or open events gated to safe zones (no releases over
edges or above non-targets)?
Are speed reductions near boundaries smooth (no oscillatory slowing
or speeding)?
Do lighting or background changes increase risky behavior (e.g.,
grazing obstacles)?
When both arms move, do inter-arm clearances stay within safe
bounds?
```

```
Robustness to Distractors:
How robust is target selection under clutter?
Do look-alikes systematically derail selection?
Is planning still collision-free amid nearby objects?
How viewpoint-sensitive is performance with clutter present?
Do purely visual (non-physical) distractors cause errors?
Are failures concentrated in specific layouts or counts?
With additional task-irrelevant objects of varying types and
counts, does the policy avoid misgrasp and confusion?
As distractors move closer to the object, can the system still
select and manipulate the correct target consistently?
When distractors share key affordances (e.g., handles), does the
policy still pick the intended target?
Do look-alike objects (color/shape/size) cause target selection
errors?
Is there a tipping point in distractor count where behavior
degrades sharply?
As distractors touch or overlap the target, can the system still
localize and act reliably?
When key target features are partially occluded, does performance
remain stable?
With multiple plausible objects present, does the policy follow the
instruction's intent (the \right" target)?
```

```
Multi-Task:
Can the policy correctly understand the affordances of the
manipulated object?
Can the policy use substitute tools to accomplish similar tasks?
```

```
Under partial occlusion, does execution across different tasks
remain successful?
What is the policy's success rate across grasp-and-place tasks?
Given different task goals for the same object, can the policy
adapt its manipulation accordingly?
In a multi-task setting, is the policy's behavior consistent under
variations in language phrasing (synonyms or simplifications)?
Across the task set, can the policy maintain a consistent
understanding of object categories and their basic affordances?
Is the understanding and execution of basic spatial relations
consistent across tasks?
```

### A.4.2 TAXONOMY

To systematically organize the queries, we derive a hierarchical taxonomy based on common concerns in robotic manipulation research. At the top level, we define five coarse categories: generalization, performance, safety, robustness, and multi-task. The multi-task category is reserved for queries that specifically target the evaluation of multi-task policies (most of which are VLAs). We further refine the taxonomy into multiple levels. For example, under "generalization" we distinguish between object generalization and scene generalization, and so on; object generalization is further split into positional generalization and appearance generalization, and so on.

### A.4.3 ANNOTATION PROTOCOL AND INTER-ANNOTATOR AGREEMENT.

For the annotation process, four graduate students or PhD students specializing in robotics participate as annotators. For each user query, annotators are asked to provide a set of sub-aspects. We then apply a majority-vote scheme to decide which sub-aspects are included in the final dataset. In cases where there is a tie, we prioritize the opinion of the annotator with more domain experience (higher seniority) as the arbiter.

### A.5 DETAILED COMPARISON BETWEEN SIMULATION BENCHMARKS AND MANIPEVALAGENT

Tab. 10 summarizes the differences between existing static simulation benchmarks and ManipEvalAgent in certain evaluation characteristics. Traditional benchmarks rely on pre-defined tasks and evaluation processes, ensuring Absolute Correctness, but they cannot adjust the evaluation content based on user queries, and their outputs are mostly limited to a single scalar success-rate score. Additionally, they lack Dynamic Generation & Evaluation and the ability to Open Tool-Use. In contrast, ManipEvalAgent drives evaluation through natural language, generates tasks and tools as needed, and supports the integration of rule-based metrics with external tools such as VLM/VQA. This enables a more flexible, analytical evaluation process while maintaining reasonable reliability.

Overall, Tab. 10 demonstrates the complementarity between ManipEvalAgent and traditional simulation benchmarks in evaluation capabilities. Both have their strengths and can play important roles in addressing different research needs. ManipEvalAgent is a complementary evaluator to traditional benchmarks, rather than a new benchmark.

### A.6 DISCUSSION ON SOME RELATED WORK AND FUTURE WORK

### A.6.1 TASK AND ASSET GENERATION WITHIN SIMULATION

Recently, several researchers have begun investigating generation tasks within simulation environments.

Some works study task scene generation in simulators (Wang et al., 2023b;c). ManipEvalAgent draws on several sound engineering practices from these works and builds a more complete framework that couples proposal, generation, and execution into an evaluation loop.

Other works go further to explore 3D asset generation within simulation (Katara et al., 2024) for creating manipulable objects. At present, ManipEvalAgent can only retrieve and use existing assets

Table 10: Comparison of Evaluation Characteristics

| Property | Existing Simulation benchmarks | ManipEvalAgent |
|---|---|---|
| User-Query Driven Evaluation | ✕ | ✓ |
| Interpretability Output | ✕ | ✓ |
| Dynamic Generation & Evaluation | ✕ | ✓ |
| Open Tool-Use | ✕ | ✓ |
| Absolute Correctness | ✓ | ✕ |

available in the simulator. We consider integrating an asset-generation pipeline into ManipEvalAgent to be a valuable direction for future work.

### A.6.2 AUTOMATING MORE STAGES OF ROBOTIC MANIPULATION RESEARCH

ManipEvalAgent can be viewed as a simulation of how an experienced human researcher evaluates manipulation policies, aiming to automate this process. In robotic manipulation research, many additional stages could be automated by multi-agent systems, including (but not limited to) analyzing user requirements, designing and implementing manipulation policies, collecting data, training policies, and iteratively refining them based on evaluation results, thereby forming a relatively closed-loop automated research workflow. Recent work (Trirat et al., 2024; Tang et al., 2025) has begun to explore automated research, providing useful references for us.

### A.6.3 SYSTEM USABILITY: PORTABILITY, STABILITY, OBSERVATIONS, AND REAL-WORLD DEPLOYMENT

Although ManipEvalAgent only needs to interact with a small set of well-defined interfaces to be deployed on different simulation engines, migrating across simulators still requires a certain amount of manual adaptation by human researchers. An interesting direction is to build a simulator-agnostic automatic evaluation system, where the evaluation process primarily operates at the GUI level, hiding the details of the underlying simulator interfaces. The rapid progress of GUI agents and coding agents supports the feasibility of this idea (Ye et al., 2025; Zhang et al., 2025a; Wang et al., 2024d; Zhang et al., 2024b).

From a longer-term perspective, a evaluation system that supports natural-language input/output and achieves behavior that is close to 100% stable could in principle subsume both current simulation benchmarks and our system in terms of evaluation capability. Thus, a valuable research direction is how to construct a unified framework that combines the stability and reproducibility of existing simulation benchmarks with the natural-language-driven, flexible evaluation capabilities of ManipEvalAgent.

The current version of ManipEvalAgent is built purely on simulation engines and relies on them to construct scenes. Some recent work (Zhou et al., 2025; Li et al., 2024b) has started to investigate how to automate evaluation of robotic manipulation policies in the real world, and these attempts provide inspiration for eventually deploying ManipEvalAgent in real world environments.

With the rapid progress of computer vision, many tools can participate in the evaluation process. Beyond RGB observations, additional visual signals can be incorporated to provide richer information and improve evaluation reliability. Compared with an RGB camera, event camera is more sensitive to high speed motion and edge changes and depends less on appearance texture, which can effectively complement the evaluation inputs (Liu et al., 2025; Gallego et al., 2020; Gehrig & Scaramuzza, 2023; Kong et al., 2025). Depth estimation, in turn, provides explicit geometric information(Yan et al., 2025; Ranftl et al., 2021; Yuan et al., 2022; Bhat et al., 2023; Yang et al., 2024).

### A.6.4 ABOUT TEST TIME ADAPTATION

Test time adaptation (Wang et al., 2022; Niu et al., 2022; Yuan et al., 2023; Boudiaf et al., 2022; Lee et al., 2024; Niu et al., 2023; Deng et al., 2025; Xu et al., 2025; Hu et al., 2025; Tan et al., 2025; Wang et al., 2025; Deng et al., 2023; Wen et al., 2023) has been widely studied across domains. It addresses

distribution shift between training and deployment and aims to improve performance under out of distribution settings. In robotic manipulation, distribution shift arises from multiple domains. ManipEvalAgent can produce diagnostic signals that localize failure modes in an interpretable way, which can trigger or guide the adaptation process.

## A.7 THE USE OF LARGE LANGUAGE MODELS (LLMS)

We used large language models to assist with language polishing and minor editorial improvements to this manuscript, including grammar, phrasing, and clarity.

