# OpenReview forum: "ManipEvalAgent: Promptable and Efficient Evaluation Framework for Robotic Manipulation Policies"
_ICLR.cc/2026/Conference — ICLR 2026 Poster_

### Official Review · Reviewer_om7e · 2025-10-30

**Soundness:** 3
**Presentation:** 3
**Contribution:** 3
**Rating:** 6
**Confidence:** 4

**Summary:**

This paper proposes ManipEvalAgent, a promptable, multi-agent framework to evaluate robotic manipulation policies. Instead of relying on fixed task suites, it turns open-ended user requests into concrete tasks and tools (via planning, code synthesis, and retrieval), runs policies in simulation, and iteratively refines the evaluation. The system mixes rule-based checks with VLM/VQA signals to produce aspect-level judgments and claims to reach similar conclusions to standard benchmarks with fewer samples and less wall-clock time.

**Strengths:**

1. Originality: Treats evaluation as an agentic, prompt-driven process; creative blend of program synthesis, retrieval, and vision tools.
2. Quality: Clear three-agent architecture; thoughtful engineering (retrieval-first, tool registry, README grounding); includes ablations and error breakdowns. I think the pipeline make sense.
3. Clarity: The decomposition into sub-aspects and the iterative probing loop are easy to follow; examples help.

**Weaknesses:**

1. Most evidence seems tied to a single simulator; claims of easy transfer aren’t yet demonstrated.
2. “Consistency of conclusions” needs formal definitions, confidence intervals, and sensitivity to the number of rollouts.
3. Reliance on VQA and generated tools raises calibration and brittleness concerns; mitigation is only partially explored.
4. Policy diversity and multi-suite coverage are modest; the open-ended query set lacks details on taxonomy and inter-annotator agreement.

**Questions:**

1. It will be better to define the agreement metric(s), report CIs, and add a trial-count sensitivity study (e.g., 3/5/10/20 rollouts) if possible.
2. Show human spot-checks vs. VQA, multi-VLM agreement, and simple perturbation tests; add unit tests for generated tools.
3. Measure human–agent agreement on sub-aspect judgments, not just aggregate consistency.
4. Do the authoer plan to opensource it? It should be clarified about the open-ended query taxonomy, labeling protocol, and inter-annotator agreement; consider releasing it?

---

> ### Author Response · Authors · 2025-11-20
> **Response to Reviewer om7e**
>
> We sincerely thank reviewer for the detailed and thoughtful feedback. Below we respond to each weakness (1–4) and question (1–4) in turn.
>
> Weakness 1：
>
> We thank the reviewer for highlighting the concern about transfer beyond a single simulator. We have added more details in App. A.3.2 on how ManipEvalAgent is ported across engines and instantiated on LIBERO. The system is deliberately designed with low coupling: to move between simulators, we only require a small set of well-defined interfaces (e.g., how to construct a task scene and how to query state during execution).
> We agree that a truly simulator-agnostic evaluation system, requiring minimal engine-specific adaptation, is an important future direction, and we now discuss this more explicitly in App. A.6.3.
>
> Weakness 2：
>
> We thank the reviewer for pointing out the incompleteness of our original agreement analysis. In App. A.2.5 and Tab. 9 we now provide a more systematic study. Instead of sampling a single random task, we adopt a multi-task setting: for each simulator we partition tasks into clusters (by horizon or by LIBERO subset) and exhaustively evaluate all tasks within each cluster, which mitigates task heterogeneity. We report policy ranking consistency via Spearman rank correlation between the five policies (ACT, Diffusion Policy, DP3, π-0, RDT-1B) under Simulation benchmark vs. ManipEvalAgent, averaged over multiple rollouts. We also report uncertainty as confidence intervals (90% CI for 10 rollouts, 95% CI for 20/50 rollouts) and explicitly study the sensitivity of agreement to the number of rollouts (10/20/50), with the seed fixed to 0 following common practice.
>
> Weakness 3：
>
> We provide additional details in App. A.3.4. If Plan Agent fails, the current attempt is terminated, logged as a failure in the history database, and planning is restarted. For scene construction, a visual self-check renders the scene; if the scene is invalid, TaskGen is asked to regenerate the task. Generated tools are guarded by a unit-test suite: if unit tests fail, it trigger tool regeneration; if unexpected exceptions occur during execution, it terminate that round, record the failure, and restart the current evaluation round.
>
> Weakness 4：
>
> App. A.4.2 and A.4.3 now describe the multi-level taxonomy of our open-ended query dataset and the annotation protocol: multiple researchers serve as annotators, and examples are included based on majority vote.
>
> Question 1:
>
> See Weakness 2
>
> Question 2:
>
> We appreciate the reviewer’s suggestion to calibrate and stress-test the VQA component. In App. A.2.4 and Tabs. 7–8, we now report a series of VQA experiments that measure both human-aligned accuracy and robustness under distribution shifts. Concretely, in RoboTwin 2.0 we construct three typical domain-randomization axes in addition to a Clean setting. We collect evaluation clips and have human researchers annotate for each clip (e.g., whether the end-effector is gradually drifting), which serve as ground truth. We then run three VLMs (GPT-4o, Gemini 1.5 Pro, GPT-4o mini) on these clips, obtain scalar VQA scores, and compute classification accuracy at a fixed threshold as well as AUROC over all thresholds. Across models and conditions, VQA performance is consistently strong; perturbations induce only mild degradation, reflecting the fact that our VQA tasks are deliberately simple and standardized, and that prompts were iteratively tuned by human experts.
>
> Regarding unit tests for generated tools, we clarify that our error breakdown for the tool-generation stage (Sec. 4.4) is in fact based on running such unit tests on the synthesized tools. We have now made this connection explicit and added a short description in App. A.3.4.
>
> Question 3:
>
> We have added an experiment in App. A.2.3 and Tab. 6 to measure the agreement between Plan Agent and human experts on sub-aspect decomposition. Results show that current Plan Agent’s sub-aspect planning is already well aligned with human experts.
>
> Question 4:
>
> Clarifications on the open-ended query taxonomy and inter-annotator agreement are provided in our response to W4.
> Our goal in this work is precisely to provide the robotic manipulation community with a novel evaluation system, so open-sourcing is a natural and essential step. We are committed to releasing our code and associated resources once the project has reached a stable and polished stage, and we expect this to happen soon.
> More broadly, the planned open-source release is part of an ambitious project that includes multiple major components, of which ManipEvalAgent is only one module. We discuss this larger framework and its additional innovations in detail in App. A.6.2.
>
> We hope these clarifications address reviewer’s concerns and help clarify the intended scope and contribution of ManipEvalAgent.

---

> ### Author Response · Authors · 2025-11-26
> **Further Response to Weaknesses 2**
>
> Thank you for your valuable feedback. This response provides a more detailed explanation for Weakness 2, addressing the concerns raised by the reviewer, and supplements our initial reply by elaborating on points that were briefly covered due to space constraints.
>
> Regarding your concern that "'Consistency of conclusions' needs formal definitions, confidence intervals, and sensitivity to the number of rollouts," we have added experiments on the consistency of conclusions between different evaluation methods.
>
>
>
> We examine how consistent ManipEvalAgent is with standard simulation benchmarks in terms of policy ranking. For each benchmark and task cluster (RoboTwin 2.0 short/medium/long horizons; LIBERO-Object/Spatial/Goal/Long), we evaluate the same five policies (ACT, Diffusion Policy, DP3, $\pi$0, and RDT-1B), aggregate the default success-rate metric over all tasks in the cluster, and derive a ranking of the five policies. We then compute Spearman's rank correlation coefficient $\rho$ between this ranking vector and the one induced by ManipEvalAgent, which focuses on the agreement in relative ordering rather than the absolute scale of scores.
>
> To quantify uncertainty, we report bootstrap confidence intervals (CI) on $\rho$. For settings with 10 rollouts per task we use 90% CIs, while for 20 and 50 rollouts we report 95% CIs. Across all clusters, we follow each benchmark's standard evaluation configuration and fix the environment seed to 0. As shown in Table, the resulting rank correlations are consistently high and stable across different rollout budgets, indicating strong agreement between ManipEvalAgent and conventional simulation-based evaluations.
>
> #### Policy ranking consistency between standard Simulation benchmarks and ManipEvalAgent
>
> | **Setting**                        | **10 rollouts**          | **20 rollouts**          | **50 rollouts**          |
> |------------------------------------|--------------------------|--------------------------|--------------------------|
> |                                    | **$\rho$ (90% CI)**       | **$\rho$ (95% CI)**       | **$\rho$ (95% CI)**       |
> | RoboTwin 2.0 (short, 0--500)           | 0.83 [0.75, 0.90]        | 0.79 [0.65, 0.90]        | 0.81 [0.70, 0.90]        |
> | RoboTwin 2.0 (medium, 600--1000)       | 0.86 [0.70, 0.90]        | 0.81 [0.60, 0.90]        | 0.83 [0.75, 0.90]        |
> | RoboTwin 2.0 (long, 1100+)             | 0.83 [0.65, 0.90]        | 0.77 [0.70, 0.90]        | 0.80 [0.65, 1.00]        |
> | LIBERO-Object                      | 0.80 [0.70, 1.00]        | 0.91 [0.70, 1.00]        | 0.82 [0.70, 0.90]        |
> | LIBERO-Spatial                     | 0.76 [0.65, 0.85]        | 0.81 [0.65, 0.90]        | 0.82 [0.60, 0.90]        |
> | LIBERO-Goal                        | 0.81 [0.70, 0.90]        | 0.73 [0.60, 0.85]        | 0.79 [0.70, 0.90]        |
> | LIBERO-Long                        | 0.85 [0.70, 0.90]        | 0.87 [0.75, 1.00]        | 0.85 [0.80, 0.90]        |
>
> These details have been added to Appendix A.2.5 and Table 9.
>
> Thank you again for your thoughtful feedback, and we look forward to your further comments.

---

> ### Author Response · Authors · 2025-11-26
> **Further Response to Question 2**
>
> Thank you for your valuable feedback. This response provides a more detailed explanation for Question 2, addressing the concerns raised by you, and supplements our initial reply by elaborating on points that were briefly covered due to space constraints.
>
> Regarding your concern, "show human spot-checks vs. VQA, multi-VLM agreement, and simple perturbation tests," in Appendix A.2.4 and Tables 7–8, we explain how VQA performs stably within our evaluation system.
>
> VQA accuracy under different perturbations on RoboTwin 2.0
>
> | **Model**         | **Clean** | **Scene Clutter** | **Background Textures** | **Lighting** |
> |-------------------|-----------|-------------------|-------------------------|--------------|
> | GPT-4o            | 0.997     | 0.989             | 0.992                   | 0.994        |
> | Gemini 1.5 Pro    | 0.998     | 0.980             | 0.987                   | 0.988        |
> | GPT-4o mini       | 0.984     | 0.984             | 0.996                   | 0.985        |
>
> VQA AUROC under different perturbations on RoboTwin 2.0
>
> | **Model**         | **Clean** | **Scene Clutter** | **Background Textures** | **Lighting** |
> |-------------------|-----------|-------------------|-------------------------|--------------|
> | GPT-4o            | 0.982     | 0.976             | 0.980                   | 0.980        |
> | Gemini 1.5 Pro    | 0.987     | 0.966             | 0.976                   | 0.981        |
> | GPT-4o mini       | 0.972     | 0.975             | 0.983                   | 0.972        |
>
> We conduct an analysis of the VQA used in our system, examining their agreement with human researcher annotations and their robustness under distribution shifts. We construct perturbation conditions along three typical domain randomization axes in RoboTwin 2.0: beyond the Clean setting, we inject task-irrelevant distractor objects on the table (Scene Clutter), with distractors sampled from RoboTwin-OD; randomize background and tabletop textures (Background Textures) by sampling from a texture library that is loaded at run time in simulation; and randomly vary lighting, including color temperature, type, number, position, and intensity. For each model, we repeat the same VQA evaluation protocol under the four settings: Clean, Scene Clutter, Background Textures, and Lighting.
>
> We collect a set of evaluation clips in the RoboTwin 2.0 environment and have human researchers annotate, in a binary manner, the key question for each clip (e.g., whether the tool gradually drifts), which we treat as VQA ground truth. We then run three VLMs (GPT-4o, Gemini 1.5 Pro, and GPT-4o mini) on the same clips and queries, and record their VQA outputs, including both natural-language descriptions and scalar scores. Based on these scores and human annotations, we compute VQA classification accuracy at a fixed threshold and AUROC over all possible thresholds. As shown in the tables above, VQA performance is overall strong, which we attribute to the fact that VQA tasks in our system are intentionally simple and consistent, and to the iteratively refined prompt engineering performed by human experts during system development. The added perturbations only cause slight degradation in VQA metrics, indicating that current VLMs already exhibit fairly robust visual capabilities in these settings.
>
> These details have been added to Appendix A.2.4 and Table 7-8
>
> Thank you again for your thoughtful feedback, and we look forward to your further comments.

---

> ### Author Response · Authors · 2025-11-26
> **Further Response to Question 4**
>
> Thank you for your valuable feedback. This response provides a more detailed explanation for Question 4, addressing the concerns raised by you, and supplements our initial reply by elaborating on points that were briefly covered due to space constraints.
>
> Regarding your concern, "do the authoer plan to opensource it? consider releasing it?", we have already clarified in our first response our firm commitment to open-sourcing and our motivation to contribute to robotics community. In addition, we further discuss the full scope of our forthcoming work in Appendix A.6.2.
>
> ManipEvalAgent can be viewed as a simulation of how an experienced human researcher evaluates manipulation policies, aiming to automate this process. In robotic manipulation research, many additional stages could be automated by multi-agent systems, including (but not limited to) analyzing user requirements, designing and implementing manipulation policies, collecting data, training policies, and iteratively refining them based on evaluation results, thereby forming a relatively closed-loop automated research workflow. Recent work [1, 2] has begun to explore automated machine learning and research, providing useful references for us.
>
> This framework is pioneering，and that you will soon see our work. These discussions have been added to Appendix A.6.2
>
> Thank you again for your thoughtful feedback, and we look forward to your further comments.
>
> [1] Trirat, P., Jeong, W., & Hwang, S. J. Automl-agent: A multi-agent LLM framework for full-pipeline AutoML. arXiv preprint arXiv:2410.02958, 2024.
> [2] Tang, J., Xia, L., Li, Z., & Huang, C. AI-Researcher: Autonomous Scientific Innovation. arXiv preprint arXiv:2505.18705, 2025.

---

### Official Review · Reviewer_YKWi · 2025-10-31

**Soundness:** 4
**Presentation:** 4
**Contribution:** 4
**Rating:** 6
**Confidence:** 3

**Summary:**

This paper presents ManipEvalAgent, a promptable, multi-round, few-sample evaluation framework for robotic manipulation policies. Instead of exhaustively running fixed benchmark suites, the system plans sub-aspects from a user query, generates tasks and evaluation tools as Python code against a simulator, executes small batches, and adapts subsequent probes based on interim observations. Tools combine rule-based metrics (via simulator APIs) and VLM-based VQA over rendered videos, and results are aggregated into interpretable textual diagnostics rather than a single score. Experiments on RoboTwin 2.0 and LIBERO suggest ManipEvalAgent reaches conclusions comparable to standard pipelines with far fewer samples; an ablation shows RAG / visual self-reflection.Agent each improve code-generation success; an error breakdown attributes most failures to TaskGen/ToolGen. Multi-task evaluation is also discussed.

**Strengths:**

Promptable, adaptive evaluation that mirrors how humans probe policies, rather than fixed suites; clear three-stage design (Proposal / Generation / Execution).

Agentic code generation for both tasks and tools, with RAG + visual self-check + README.Agent to stabilize generation—well engineered and ablated.

Hybrid metrics (rule-based + VQA) let the evaluator capture aspects not exposed by simulator APIs, enabling finer-grained diagnostics.

Evidence of agreement with standard benchmarks on several dimensions (success rate and LIBERO sub-suites), while using fewer samples; multi-task variant also reported.

Failure analysis reveals where the system breaks (generation stage dominates), which is actionable for future iterations.

**Weaknesses:**

Agreement protocol is under-specified / potentially fragile. Appendix A.2.1 defines agreement by comparing a single randomly chosen task per benchmark against 10 runs of the agent and checking whether the benchmark SR lies within 1σ/3σ of the agent’s mean. This ignores task heterogeneity, seed variance, and policy × task interactions, risking unstable conclusions from small-N sampling. Please justify the statistical validity and add per-aspect calibration beyond one random task.

Ground-truthing of VQA metrics is not calibrated. VLM-based tools produce numeric judgments, but there is no report of AUROC/ECE, threshold selection, or prompt sensitivity under distribution shift (lighting, gloss, clutter). Given VQA results feed aggregation and next-round planning, lack of calibration can systematically steer the loop.

**Questions:**

See weaknesses

---

> ### Author Response · Authors · 2025-11-20
> **Response to Reviewer YKWi**
>
> We sincerely thank reviewer for the detailed and thoughtful feedback. Below we respond to each weakness (1–2).
>
> Weakness 1:
>
> We thank the reviewer for pointing out the incompleteness of our original agreement analysis. In App. A.2.5 and Tab. 9 we now provide a more systematic study. Instead of sampling a single random task, we adopt a multi-task setting: for each simulator we partition tasks into clusters (by horizon or by LIBERO subset) and exhaustively evaluate all tasks within each cluster, which mitigates task heterogeneity. We report policy ranking consistency via Spearman rank correlation between the five policies (ACT, Diffusion Policy, DP3, π-0, RDT-1B) under Simulation benchmark vs. ManipEvalAgent, averaged over multiple rollouts. We also report uncertainty as confidence intervals (90% CI for 10 rollouts, 95% CI for 20/50 rollouts) and explicitly study the sensitivity of agreement to the number of rollouts (10/20/50), with the seed fixed to 0 following common practice. These additions make our definition of agreement and its trial-count sensitivity more formal and transparent.
>
> Weakness 2:
>
> We appreciate the reviewer’s suggestion to calibrate and stress-test the VQA component. In App. A.2.4 and Tabs. 7–8, we now report a series of VQA experiments that measure both human-aligned accuracy and robustness under distribution shifts. Concretely, in RoboTwin 2.0 we construct three typical domain-randomization axes in addition to a Clean setting: (i) Scene Clutter, where task-irrelevant objects sampled from RoboTwin-OD are added; (ii) Background Textures, where table and background textures are randomized from a texture library; and (iii) Lighting, where color temperature, type, number, position, and intensity of lights are randomized. We collect evaluation clips and have human researchers annotate for each clip (e.g., whether the end-effector is gradually drifting), which serve as ground truth. We then run three VLMs (GPT-4o, Gemini 1.5 Pro, GPT-4o mini) on these clips, obtain scalar VQA scores, and compute classification accuracy at a fixed threshold as well as AUROC over all thresholds. Across models and conditions, VQA performance is consistently strong; perturbations induce only mild degradation, reflecting the fact that our VQA tasks are deliberately simple and standardized, and that prompts were iteratively tuned by human experts.
>
> We hope these clarifications address reviewer’s concerns and help clarify the intended scope and contribution of ManipEvalAgent.

---

> > ### Comment · Reviewer_YKWi · 2025-11-20
> > **Response to authors**
> >
> > Dear Authors,
> >
> > Thank you for your efforts to address weakness and answer my questions. I choose to update my score to 8 and confidence to 4. Hope your paper will be accepted.

---

> > > ### Author Response · Authors · 2025-11-21
> > > **Response to Reviewer YKWi**
> > >
> > > Thank you for taking the time to carefully read our response; we feel truly honored.
> > > We will continue to refine this work, with the goal of providing the manipulation community with a useful tool.
> > > We are grateful for your constructive feedback and encouragement, and wish you all the best.

---

### Official Review · Reviewer_8VKw · 2025-11-02

**Soundness:** 2
**Presentation:** 2
**Contribution:** 2
**Rating:** 2
**Confidence:** 4

**Summary:**

This paper proposes a simulation benchmark for robotic manipulation that is adaptable and agentic. Users can ask specific questions about how they want their policy evaluated, like "how robust is my policy to variations in object colors?", and the simulation engine will use AI models to piece together simulation assets and call simulation APIs to construct a series of scenes that test the user's query. The simulation engine analyzes results evaluating a policy on a scene it created, and then based on the results decides what scene to construct next for evaluation. The main motivation the authors propose for having such an agentic evaluation platform is that simulated evaluations take quite a long time to run, and at the end you usually just get back one numerical score specifying how well your policy did, and so it's hard to debug what exactly the policy does/doesn't do well. In evaluating their AI-based evaluation platform, the authors find that indeed time can be saved, with baseline simulated evaluation platforms taking 2+ hours to evaluate, but the proposed approach taking ~45 minutes. Rankings of policies on the proposed adaptable benchmark also correlate with rankings on previous widely used simulation benchmarks.

**Strengths:**

(1) It is often the case that researchers want to learn what a policy they've trained does well at, and what it fails at, and there can be a lot of nuance in where the policy fails/does well. At the same time existing evaluation platforms usually just output a single numerical score, making it hard for the researcher to get this more nuanced insight into the performance of their policy. This paper proposes an approach by which the evaluation platform can respond to user queries and adaptively build a set of tasks to benchmark on that will answer the question.

(2) The evaluation system seems well engineered and complete from the paper's description of it.

(3) Simulated scenes/assets usually fall prey to difficulties of scale, spurring recent work in procedural generation. This paper's evaluation platform can viewed as another instance of procedural generation, which makes it more scalable.

**Weaknesses:**

(1) While the proposed simulation benchmark can foreseeably be useful to researchers for model development and debugging, from my understanding it will be difficult to use it as a *benchmark*. Because the benchmark is not static, and the scenes/tasks change based on user queries, it becomes difficult to compare various methods objectively. Further, even if the same prompt is used (i.e., "evaluate how good my policy is on pick and place tasks") the AI-powered evaluation platform may be non-deterministic and thus generate different scenes for different policies, again preventing comparisons.

(2) While the idea is interesting, the premise that a new simulation evaluation platform should be built at all is in my opinion problematic. There are several simulation benchmarks for robotic manipulation, and much work has tried to develop robotic policies that push the frontier of these benchmarks. However there is no guarantee that good performance on these simulation benchmarks translates to good real world robotic performance (which generally is what the robotics community is aiming for), and so a new simulation benchmark misses the forest for the trees, and promotes the optimization of a metric that does not necessarily have great correlation with the fundamental scientific questions researchers are interested in answering -- how well do policies do on real-world tasks. In my opinion there is still room for simulated evaluation benchmarks, like hyper-realistic benchmarks that very accurately model real world physics/visuals, but this work does not tackle that problem.

(3) Table 2 shows how well the ranking of policies produced by the proposed benchmark correlate with rankings of the same policies on previous simulation benchmarks. The correlation does not seem to be very strong, and even if it was, correlation with prior simulations shouldn't even be a metric to optimize, because there is no evidence that those prior simulations correlate with the real world.

(4) The motivation that time can be saved with the proposed benchmark is not a great motivation. Simulation evaluation platforms are generally considered to be cheap (relative to real-world evaluations, hence their appeal), and a drop in evaluation time from 2 hours to 45 minutes isn't very substantial.

**Questions:**

(1) How can the proposed agentic simulated evaluation system be used as a benchmark, which as far as I understand is the goal of the work (see weakness (1))?

(2) Table 3 shows how often the evaluation platform fails to evaluate the policy (e.g., due to AI issues). While it doesn't fail often, which is good, when it does fail what should the user do to get their policy evaluated?

(3) The evaluation makes use of VQA models in various parts -- what happens if the VQA model makes mistakes? How does this affect the set up of the evaluation and the results? Can the results still be useful to researchers if the VQA model can make mistakes?

---

> ### Author Response · Authors · 2025-11-20
> **Response to Reviewer 8VKw.**
>
> We sincerely thank reviewer for the detailed and thoughtful feedback. Below we respond to each weakness (1–4) and question (1–3) in turn.
>
> Weakness 1:
>
> We thank the reviewer for raising this concern and apologize for the confusion caused by our terminology. We clarify that ManipEvalAgent is not proposed as a new standalone benchmark, but as a complementary evaluation system on top of existing simulation benchmarks, adding capabilities that current hand-designed suites lack: user-query–driven evaluation, interpretable textual output, dynamic generation & probing, and open tool use, whereas standard benchmarks focus on absolutely correct, fixed task suites. We now make this distinction explicit in App. A.5 and Tab.10.
> We agree that this points to an important future direction, as discussed in App. A.6.3.
> Regarding comparability across queries, policies, and runs, App. A.3.3 details several design choices: a history database of past evaluations that is retrieved into the Plan Agent to reuse successful plans for similar queries. All other retrieval-augmented generation modules are handled in the same way, which further promotes consistency across queries, policies, and evaluation runs. These mechanisms constrain variability across evaluation instances; we regret not explaining them more clearly in the main text.
>
> Weakness 2:
>
> We respectfully disagree with the concern regarding sim-to-real considerations. The issue you raise is fundamentally a sim-to-real generalization problem, which we see as orthogonal to our research question. Our focus is on designing a promptable, adaptively planned, and interpretable evaluation system on top of existing simulators. We therefore believe that this does not diminish the significance of our contribution.
> We agree, however, that a unified evaluation framework that can operate both in simulation and on real-world robots is an important future direction. We now explicitly discuss this perspective and future direction in App. A.6.3.
>
> Weakness 3:
>
> We respectfully disagree with the concern regarding sim-to-real considerations, for reasons related to W2.
> We have added a more complete analysis in App. A.2.5 and Tab. 9, where we report policy ranking consistency (Spearman rank correlation with confidence intervals). These results show that ManipEvalAgent achieves reasonably strong and stable agreement with current simulation benchmarks. We believe such consistency is meaningful: it demonstrates that ManipEvalAgent not only introduces new capabilities (promptable, adaptive, interpretable evaluation), but also does not substantially drift away from what existing simulators already assess.
>
> Weakness 4:
>
> Embodied manipulation research, especially on VLA, typically involves large evaluation suites, where a full simulation pass can take days. In our multi-task setting, ManipEvalAgent reduces end-to-end evaluation from days to hours, which we consider a substantial gain in practical research iteration speed; the corresponding results are reported in App. A.2.2 and Tab. 4. We apologize that, due to ICLR space constraints, these experiments were moved to the appendix and may have been easy to miss.
>
> Question 1:
>
> See Weakness 1
>
> Question 2:
>
> We apologize for not clearly describing our failure-handling mechanisms. We provide additional details in App. A.3.4. If Plan Agent fails, the current attempt is terminated, and planning is restarted. For scene construction, a visual self-check renders the scene; if the scene is invalid, TaskGen is asked to regenerate the task. Generated tools are guarded by a unit-test suite: if unit tests fail, it trigger tool regeneration.
> We acknowledge that we did not spell out these (otherwise straightforward) engineering choices clearly enough.
>
> Question 3:
>
> We agree that VQA reliability is important and have added a more detailed analysis in App. A.2.4 (Tabs. 7–8). There we show that, in our setting, VQA error rate is low: VLM judgments exhibit strong cross-model agreement and remain robust under simple distribution shifts, largely because we assign VQA only very simple, standardized queries and iteratively optimized the prompts during system development.
> We clarify the data flow for extracting evaluation-relevant information from the simulator in App. A.3.5. Python tools first read state directly from the simulator; for aspects that are more naturally judged from video, it construct concise natural-language questions and feed them to the VLM together with tool outputs and keyframes. All signals (rule-based + VQA) are then aggregated by the Plan Agent across multiple trials, which makes the system reasonably robust to occasional VQA mispredictions given the overall low error rate.
>
> We hope these clarifications address reviewer’s concerns and help clarify the intended scope and contribution of ManipEvalAgent.

---

> ### Author Response · Authors · 2025-11-25
> **Further Response to Weaknesses 1**
>
> Thank you for your valuable feedback. This response provides a more detailed explanation for Weakness 1, addressing the concerns raised by the reviewer, and supplements our initial reply by elaborating on points that were briefly covered due to space constraints.
>
> Regarding your concern that "even if the same prompt is used, the AI-powered evaluation platform may be non-deterministic and thus generate different scenes for different policies, again preventing comparisons," we have added a detailed description of the mechanisms our system uses to maintain consistency across queries.
>
> To maintain stability and cross-query consistency across different evaluation instances, we adopt a few simple design choices. First, in proposal stage, we maintain a historical evaluation database that stores, for each past evaluation, the corresponding planning information, including the original user query and its sub-aspects. When a new user query arrives, system retrieves similar past queries from this repository and injects their planning results into Plan Agent’s context, allowing the agent to reuse existing decompositions for similar problems. This simple yet effective mechanism significantly improves the stability and behavioral consistency of cross-query and cross-policy evaluations.
>
> Generation stage, both for task generation and tool generation, relies on retrieval-augmented generation (RAG): agents retrieve code snippets and other relevant artifacts from dedicated task and tool repositories that best match the current query and its sub-aspects, and then condition on these retrieved materials during generation. Since similar queries tend to trigger similar retrieval results, this mechanism further preserves consistency across repeated evaluations, causing the system to preferentially reuse a nearly identical family of task and tool definitions when evaluating different policies, thereby yielding more stable and comparable evaluation behavior.
>
> We apologize for not mentioning the engineering details in our initial submission due to space constraints, as our system is quite large. The aforementioned content has been added to Appendix A.3.3.
>
> Thank you again for your thoughtful feedback, and we look forward to your further comments.

---

> ### Author Response · Authors · 2025-11-25
> **Further Response to Weaknesses 3**
>
> Thank you for your valuable feedback. This response provides a more detailed explanation for Weakness 3, addressing the concerns raised by the reviewer, and supplements our initial reply by elaborating on points that were briefly covered due to space constraints.
>
> Regarding your concern that "The correlation does not seem to be very strong," we have added experiments on the consistency of conclusions between different evaluation methods.
>
> We further examine how consistent ManipEvalAgent is with standard simulation benchmarks in terms of policy ranking. For each benchmark and task cluster (RoboTwin 2.0 short/medium/long horizons; LIBERO-Object/Spatial/Goal/Long), we evaluate the same five policies (ACT, Diffusion Policy, DP3, $\pi$0, and RDT-1B), aggregate the default metric over all tasks in the cluster, and derive a ranking of the five policies. We then compute Spearman's rank correlation coefficient $\rho$ between this ranking vector and the one induced by ManipEvalAgent, which focuses on the agreement in relative ordering rather than the absolute scale of scores.
>
> To quantify uncertainty, we report bootstrap confidence intervals (CI) on $\rho$. For settings with 10 rollouts per task we use 90% CIs, while for 20 and 50 rollouts we report 95% CIs. Across all clusters, we follow each benchmark's standard evaluation configuration and fix the environment seed to 0. As shown in Table, the resulting rank correlations are consistently high and stable across different rollout budgets, indicating strong agreement between ManipEvalAgent and conventional simulation-based evaluations.
>
> #### Policy ranking consistency between standard Simulation benchmarks and ManipEvalAgent
>
> | **Setting**                        | **10 rollouts**          | **20 rollouts**          | **50 rollouts**          |
> |------------------------------------|--------------------------|--------------------------|--------------------------|
> |                                    | **$\rho$ (90% CI)**       | **$\rho$ (95% CI)**       | **$\rho$ (95% CI)**       |
> | RoboTwin 2.0 (short, 0--500)           | 0.83 [0.75, 0.90]        | 0.79 [0.65, 0.90]        | 0.81 [0.70, 0.90]        |
> | RoboTwin 2.0 (medium, 600--1000)       | 0.86 [0.70, 0.90]        | 0.81 [0.60, 0.90]        | 0.83 [0.75, 0.90]        |
> | RoboTwin 2.0 (long, 1100+)             | 0.83 [0.65, 0.90]        | 0.77 [0.70, 0.90]        | 0.80 [0.65, 1.00]        |
> | LIBERO-Object                      | 0.80 [0.70, 1.00]        | 0.91 [0.70, 1.00]        | 0.82 [0.70, 0.90]        |
> | LIBERO-Spatial                     | 0.76 [0.65, 0.85]        | 0.81 [0.65, 0.90]        | 0.82 [0.60, 0.90]        |
> | LIBERO-Goal                        | 0.81 [0.70, 0.90]        | 0.73 [0.60, 0.85]        | 0.79 [0.70, 0.90]        |
> | LIBERO-Long                        | 0.85 [0.70, 0.90]        | 0.87 [0.75, 1.00]        | 0.85 [0.80, 0.90]        |
>
> These details have been added to Appendix A.2.5 and Table 9.
>
> Thank you again for your thoughtful feedback, and we look forward to your further comments.

---

> ### Author Response · Authors · 2025-11-25
> **Further Response to Weaknesses 4**
>
> Thank you for your valuable feedback. This response provides a more detailed explanation for Weakness 4, addressing the concerns raised by the reviewer, and supplements our initial reply by elaborating on points that were briefly covered due to space constraints.
>
> In the initial version of the paper submission, we had already provided the evaluation time test under the multi-task setting in Table 4 in appendix. ManipEvalAgent reduces end-to-end evaluation from days to hours. Due to the page constraints of ICLR, this experimental result was moved to the appendix in the initial submitted version, and we sincerely apologize for any inconvenience this may have caused in your reading.
>
> Total evaluation time and sample count of ManipEvalAgent under the multi-task setting
>
> | **Models** | **RoboTwin**            | **LIBERO**            | **Ours**             |
> |------------|-------------------------|-----------------------|----------------------|
> | RDT        | 11102 min, 2763623 samples | 4909 min, 1073428 samples | 97 min, 27037 samples |
> | $\pi_0$    | 7900 min, 2445809 samples  | 4330 min, 992816 samples  | 68 min, 21745 samples |
>
> Under multi-task setting, traditional benchmarks require repeated runs (e.g., 100 times) across all predefined task suites to obtain an overall impression of the VLA performance. In contrast, ManipEvalAgent only requires a small number of evaluations on representative tasks, dynamically performed, to obtain the same overall impression of the VLA performance.
>
> The time reduction comes primarily from needing far fewer executions. Human researchers routinely form reliable judgments about a policy’s strengths and weaknesses by closely inspecting a handful of rollouts, rather than running 100 trials just to estimate a scalar success rate. ManipEvalAgent explicitly mimics this mode of evaluation: it extracts richer information from each execution and plans subsequent probes adaptively, enabling much more sample-efficient (and thus time-efficient) evaluation.
>
> Thank you again for your thoughtful feedback, and we look forward to your further comments.

---

> ### Author Response · Authors · 2025-11-25
> **Further Response to Question 2**
>
> Thank you for your valuable feedback. This response provides a more detailed explanation for Question 2, addressing the concerns raised by the reviewer, and supplements our initial reply by elaborating on points that were briefly covered due to space constraints.
>
> Regarding your concern, "when it does fail, what should the user do to get their policy evaluated," we would like to clarify that when any part of the evaluation system fails, there are corresponding handling mechanisms in place, and it does not result in a direct failure. We apologize for not including these engineering implementation details in the initial version, and for any confusion this may have caused.
>
> ManipEvalAgent inevitably encounters various types of failures. To prevent these failures from disrupting the overall evaluation process, we design dedicated handling mechanisms for each stage. First, in the planning stage, if the set of sub-aspects produced by the Plan Agent disagrees with the human-annotated ground truth from the open-ended query dataset on more than half of the elements, we classify it as a planning failure: the current planning attempt is terminated, the case is logged as a failure in database, and a new planning attempt is started.
>
> Second, in task generation, we employ a visual self-check to inspect the rendered scenes. If the scene is detected to be inconsistent with the intended sub-aspects or exhibits clearly abnormal object configurations, it is treated as a task-generation failure, and TaskGen Agents is required to regenerate the scene.
>
> For tool generation, we employ a unit test suite to validate the generated tool functions: if the unit tests fail, it trigger regeneration; if unexpected exceptions occur during execution, it terminate current evaluation round, record the failure, and restart that evaluation round.
>
> As for issues originating from the simulation engine itself, such failures also occur in traditional simulation benchmarks; we likewise treat them uniformly as policy execution failures. These mechanisms ensure that the system maintains overall robustness and usability even in the presence of localized failures.
>
> These details have been added to Appendix A.3.4
>
> Thank you again for your thoughtful feedback, and we look forward to your further comments.

---

> ### Author Response · Authors · 2025-11-25
> **Further Response to Question 3**
>
> Thank you for your valuable feedback. This response provides a more detailed explanation for Question 3, addressing the concerns raised by the reviewer, and supplements our initial reply by elaborating on points that were briefly covered due to space constraints.
>
> Regarding your concern, "what happens if the VQA model makes mistakes?" we have described in our first response and in Appendix A.3.5 how the plan agent handles occasional VQA errors. Additionally, in Appendix A.2.4 and Tables 7–8, we explain how VQA performs stably within our evaluation system.
>
> VQA accuracy under different perturbations on RoboTwin 2.0
>
> | **Model**         | **Clean** | **Scene Clutter** | **Background Textures** | **Lighting** |
> |-------------------|-----------|-------------------|-------------------------|--------------|
> | GPT-4o            | 0.997     | 0.989             | 0.992                   | 0.994        |
> | Gemini 1.5 Pro    | 0.998     | 0.980             | 0.987                   | 0.988        |
> | GPT-4o mini       | 0.984     | 0.984             | 0.996                   | 0.985        |
>
> VQA AUROC under different perturbations on RoboTwin 2.0
>
> | **Model**         | **Clean** | **Scene Clutter** | **Background Textures** | **Lighting** |
> |-------------------|-----------|-------------------|-------------------------|--------------|
> | GPT-4o            | 0.982     | 0.976             | 0.980                   | 0.980        |
> | Gemini 1.5 Pro    | 0.987     | 0.966             | 0.976                   | 0.981        |
> | GPT-4o mini       | 0.972     | 0.975             | 0.983                   | 0.972        |
>
> We conduct an analysis of the VQA used in our system, examining their agreement with human researcher annotations and their robustness under distribution shifts. We construct perturbation conditions along three typical domain randomization axes in RoboTwin 2.0: beyond the Clean setting, we inject task-irrelevant distractor objects on the table (Scene Clutter), with distractors sampled from RoboTwin-OD; randomize background and tabletop textures (Background Textures) by sampling from a texture library that is loaded at run time in simulation; and randomly vary lighting, including color temperature, type, number, position, and intensity. For each model, we repeat the same VQA evaluation protocol under the four settings: Clean, Scene Clutter, Background Textures, and Lighting.
>
> We collect a set of evaluation clips in the RoboTwin 2.0 environment and have human researchers annotate, in a binary manner, the key question for each clip (e.g., whether the tool gradually drifts), which we treat as VQA ground truth. We then run three VLMs (GPT-4o, Gemini 1.5 Pro, and GPT-4o mini) on the same clips and queries, and record their VQA outputs, including both natural-language descriptions and scalar scores. Based on these scores and human annotations, we compute VQA classification accuracy at a fixed threshold and AUROC over all possible thresholds. As shown in the tables above, VQA performance is overall strong, which we attribute to the fact that VQA tasks in our system are intentionally simple and consistent, and to the iteratively refined prompt engineering performed by human experts during system development. The added perturbations only cause slight degradation in VQA metrics, indicating that current VLMs already exhibit fairly robust visual capabilities in these settings.
>
> These details have been added to Appendix A.2.4 and Table 7-8
>
> Thank you again for your thoughtful feedback, and we look forward to your further comments.

---

> > ### Comment · Reviewer_8VKw · 2025-11-26
> >
> > I appreciate the authors' effort into addressing my concerns.
> >
> > Overall, weaknesses (1), (3), and (4), and questions (1), (2), and (3) have been adequately addressed by the authors.
> >
> > It is evident that the authors have thought deeply about their work. Overall the execution of the paper is very strong, and I appreciate the inclusion of various appendix sections that address my concerns.
> >
> > The one weakness that does remain is not in the execution (which is strong) but rather in the idea behind the project itself. While the idea of a promptable evaluation framework might intuitively make sense, it remains difficult (albeit not impossible) to envision how such an idea might provide substantial utility to researchers working on real-world robotics problems. The idea is currently instantiated on simulated robotics benchmarks, which are useful during early iterations of research but not for final model evaluations, which should be done in the real-world. Nonetheless, I do see how the proposed promptable evaluation idea might be instantiated on hyper-realistic sims or AutoEval style setups, and so after the authors' rebuttal I am more convinced that the proposed idea might be useful for robotics research.
> >
> > Taking everything into account, I will raise my score to a 6.

---

> > > ### Author Response · Authors · 2025-11-27
> > > **Response to Reviewer 8VKw.**
> > >
> > > Thank you for taking the time to carefully read our response; we are truly honored.
> > >
> > > We greatly appreciate your detailed and constructive feedback, which has helped us improve this work. We will continue to optimize this project with the goal of providing a practical tool for robotic community.
> > >
> > > Your insights are correct. The goal of robotic community is to achieve high-performance robots in the real world, and we will take this as a key premise for our future research.
> > >
> > > Thank you for your support and encouragement. Wishing you all the best!

---

### Author Response · Authors · 2025-11-29
**General Response and Summary of Rebuttal Process So Far**

Dear Area Chairs and Reviewers,

Thank you for your constructive feedback and for your efforts in helping us improve this work. Regarding the unexpected incident on November 27, we support the decisions made by ICLR in response, which we believe are instrumental in upholding fairness and academic integrity in the community.

To assist with your evaluation, we provide a brief summary of our work, the reviews and discussion so far, and the changes we have made during the rebuttal stage.

###  Paper Summary

- We propose a promptable, multi-round, few-sample evaluation framework for robotic manipulation policies. Instead of exhaustively running fixed benchmark suites, the system decomposes a user query into sub-aspects, generates tasks and evaluation tools as Python code against the simulator, executes small batches of rollouts, and adapts subsequent probes based on interim observations, aggregating the results into interpretable textual diagnostics rather than a single score.

- Our approach offers three key advantages: (1) efficiency, removing the need for massive sampling; (2) promptability, planning the evaluation process according to user queries; and (3) interpretability, providing diagnostic text that goes beyond a single scalar score.

- Across multiple settings, our evaluation method significantly shortens the overall evaluation time compared with traditional simulation benchmarks, while reaching conclusions comparable to those obtained from large-scale simulation benchmarks.

###  Review Status Before Author Response

Before the discussion period , we received three reviews (scores: 6, 6, 2).

Across reviewers, the consensus is:

- **Novel and practical**: a promptable, adaptive framework that responds to user queries and automatically constructs a suite of tasks, enabling efficient evaluation that closely mirrors how human researchers probe policy behavior.

- **Well designed and thoroughly engineered**: it features a clear modular architecture and a coherent end-to-end pipeline; the implementation is robust, thoughtfully engineered, and extensible to new settings.

- **Comprehensive** experimental evaluation

Reviewers raised several constructive questions, primarily concerning (i) whether VQA component is sufficiently stable within our system, and (ii) the need for further validation of the consistency between ManipEvalAgent’s evaluations and existing benchmarks.

###  Author Response and Review Status After Author Response

We have thoroughly addressed all questions raised by the reviewers and, prior to November 26 (UTC), continuously incorporated additional material into the paper PDF to respond to their concerns.

 The additions and clarifications include, but are not limited to:

- Additional experiments, including further studies on the consistency of conclusions across evaluation methods (Spearman rank correlation, confidence intervals (CI), and sensitivity to the number of rollouts), the accuracy and robustness of the VQA module, and the agreement between our aspect decomposition and human annotations.

- More detailed descriptions of the pipeline, covering the mechanisms used to maintain consistency across queries, the failure-handling procedures, and the data flow related to VQA.

- Clarifications of experimental settings, for example how error breakdown statistics are computed.

- Clarifications of the dataset taxonomy and labeling protocol.

- Additional implementation details on LIBERO

- A clearer explanation of the distinction between ManipEvalAgent and existing simulation benchmarks.

After receiving our response, multiple reviewers confirmed that their concerns had been resolved:

- Reviewer 8VKw considered 6 out of the 7 concerns to be resolved and stated that the sim-to-real concerns do not undermine the core contribution of the work, raising the score **from 2 to 6**.

- Reviewer YKWi regarded all concerns as satisfactorily addressed, increased the score **from 6 to 8** and the confidence from 3 to 4.

After a productive discussion, we received scores of **6, 8, 6**, with an **average** of **6.67** and an average confidence of 4, reflecting clear and consistent recognition from the reviewers.

The last score increase occurred at approximately **22:47 UTC on November 26**, well **before** the leak incident became widespread (around **15:00 UTC on November 27**).

Our work is aimed at providing a useful tool for the robotic manipulation community.

Thank you once again for your time and thoughtful consideration.

Sincerely,

Authors

---

### Meta-Review · Area_Chair_dWix · 2025-12-22

**Summary:**

The submission introduces a simulation-based evaluation framework for robotic manipulation. Reviewers liked the idea but are concerned about the limited evaluation, tasks, and metric consistencies, as well as the module design in the pipeline.

**Reviewer Concerns:**

Reviewers agreed that most concerns are addressed satisfactorily.

**Reviewer Scores:**

Most likely, the final scores will be 6, 8, 6.

---

### Decision · Program_Chairs · 2026-01-26

Accept (Poster)